# How a Cooperative-Override Circuit Suppresses Nash Play in Large Language Models

## Abstract

LLM agents are known to deviate from Nash equilibria in strategic interactions, but there is no causal, equilibrium-benchmarked account of the internal computation behind this, nor a demonstration that it can be reversed. We do both. Working with four open-source models (Llama-3 and Qwen2.5, 8B to 72B parameters) playing four canonical two-player games, we first establish the behavioral picture through self-play and cross-play, then open up the 32-layer Llama-3-8B model. Behaviorally, we find a scale-dependent pattern: under direct prompting the larger models (at or above 32B) fail to play Nash and lock at full cooperation in Prisoner's Dilemma, whereas Llama-3-8B plays close to Nash; chain-of-thought moves the larger models toward Nash but with substantial seed variance, and does not worsen the 8B. In cross-play, a single Nash-playing small model makes Nash play contagious in mixed pairings, collapsing the cooperation of any larger partner, while two larger models reinforce each other's cooperation indefinitely (an effect that is symmetric in role order in the clean direct condition). Mechanistically, on contrastive prompts a clean linear probe decodes the opponent's action from the residual stream perfectly at every layer, while the logit lens shows that the model favors the Nash action through most of its depth; a prosocial override (a cooperative, other-regarding bias rooted in pretraining on human text and modulated by RLHF) surges in the late layers, reaching about 80% probability of cooperation at layer 30, before the final layer commits back toward the Nash action. The 8B thus computes a cooperative override that its own final layer overrides in turn, consistent with its playing Nash; the larger models cooperate behaviorally despite a broadly similar final-layer readout, a gap we discuss in Section 7.3. This override is a linear, causally controllable direction in the residual stream: injecting it shifts behavior bidirectionally, confirmed through concept clamping. The logit lens on Llama-3-70B and Qwen2.5-72B shows the override is present across both scales and architectures, though causal intervention experiments are established only for the 8B.

large language models, activation steering, multi-agent systems, opponent modeling

## 1 Introduction

Twenty-five years ago, Broder et al. (2000) observed that the World Wide Web had grown so large and complex that it could no longer be understood through analytical derivation alone. It had become the first human artifact that demanded an empirical science of its own structure: to learn what it was, one had to observe and experiment rather than reason from first principles. Large language models occupy an analogous position today. With hundreds of billions of learned parameters and behaviors that emerge from pretraining rather than explicit design, they resist analytical study in the same way. The field of mechanistic interpretability exists precisely to bring observational methods to these artifacts. This paper applies those methods to a concrete and consequential question about LLMs in strategic settings.

A growing body of empirical work shows that LLM agents fail to converge to Nash equilibria in repeated strategic games (Brookins and DeBacker, 2024; Akata et al., 2025; Fontana et al., 2025; Jia et al., 2025). These deviations are systematic, shaped by game structure, prompting style, and model family, and qualitatively

resemble the human behavioral biases documented in experimental economics (Camerer and Ho, 1999; Goeree and Holt, 2001). At the aggregate level the behavioral question is largely settled: LLMs often fail to play Nash. As we show, however, this is uneven: the specific 8B model we open up reliably plays the Nash equilibrium in Prisoner's Dilemma (with the caveat, discussed later, that its prompt states the equilibrium), while the larger models do not, which is what makes an internal, mechanistic account necessary rather than a purely behavioral one.

The mechanistic question of *why* this is so, benchmarked against exact equilibria, and whether it is causally reversible, has received little attention; prior interpretability work on related social behaviors (Section 2) does not target equilibrium play in repeated games or establish causal control of it. This is the question we answer.

The gap matters for two reasons. Without a causal account, behavioral observations are difficult to act on: one cannot know whether prompting, fine-tuning, or architectural changes will fix the deviation or merely mask it. And if LLMs encode Nash-relevant representations that are suppressed rather than absent, that is a fundamentally different diagnosis from a model that lacks strategic competence. We show the former is true.

The closest mechanistic precedent probes small transformers trained specifically on a synthetic game (Othello), recovering board-state representations that are causal with respect to move prediction (Li et al., 2023). Our setting differs in every important dimension: we apply mechanistic tools to general-purpose instruction-tuned LLMs playing repeated strategic games, where any strategic competence must emerge from pretraining rather than game-specific training.

**Contributions.** This paper makes four contributions. We show through a clean, single-provenance linear probe that on contrastive histories the opponent's action is perfectly separable in the residual stream at every layer (a near-trivial ceiling we do not over-read; Section 7.1). We use the logit lens to show that the model internally favors the Nash action through most of its forward pass, and that a late-layer prosocial override surges toward cooperation before the final layer resolves back toward the Nash action; we characterize this circuit through attention-head zero-ablation and activation steering. We extract a Nash direction from the residual stream and demonstrate reliable, graded causal control over equilibrium play: steering toward Nash achieves 99.2% defection in Prisoner's Dilemma, and concept clamping confirms the direction is causal rather than a spurious residual feature. We run cross-play experiments across all 12 ordered heterogeneous pairings of four models and show that Nash outcomes depend on population composition: a small model breaks cooperative locks by defecting early and inducing partners to follow, two large models reinforce each other's cooperative prior in direct prompting. In the clean Direct condition these effects are symmetric in role order; apparent order effects appear only in the higher-variance reasoning conditions.

Behavioral self-play results across four models and three reasoning conditions appear in Section 5 as context for the mechanistic account.

## 2 Related Work

This paper connects three bodies of literature: behavioral studies of LLMs in strategic settings, mechanistic interpretability of transformer models, and activation steering as a tool for causal intervention.

### 2.1 LLMs in Strategic Settings

Several papers establish the behavioral baseline this work builds on. Brookins and DeBacker (2024) find that GPT-4 shows human-like cooperation in prisoner's dilemmas, which they attribute to RLHF-induced prosocial preferences. Akata et al. (2025) show LLMs can sustain cooperation in repeated games with sufficient context. Fontana et al. (2025) study three LLMs playing the iterated Prisoner's Dilemma against opponents with varying hostility levels, finding that Llama2 and GPT-3.5 cooperate more than typical humans and are especially forgiving, while Llama3 behaves more like a human player. Jia et al. (2025) evaluate 22 LLMs using a behavioral game-theoretic framework and find that chain-of-thought prompting improves but does not guarantee Nash approximation, and that model scale alone does not determine performance. We treat these findings as a starting point and focus on the mechanistic question they leave open.

## 2.2 Mechanistic Interpretability and Activation Steering

Mechanistic interpretability seeks to identify the algorithms implemented by neural networks (Olah et al., 2020; Elhage et al., 2021). Meng et al. (2022) show that factual associations are localized to specific MLP (multi-layer perceptron) layers. Attention head analysis has characterized heads for indirect object identification (Wang et al., 2023) and copy suppression (McDougall et al., 2023). The logit lens (Nostalgebraist, 2020), formalized by Belrose et al. (2023), tracks the model's evolving token prediction by projecting each layer's hidden state through the output embedding matrix. The most relevant structural precedent is Li et al. (2023), who probe a small GPT model trained on Othello sequences and recover causal board-state representations. We extend this to general-purpose instruction-tuned LLMs: no game-specific training is used, the games involve repeated interaction rather than perfect-information board states, and we go beyond probing to causal steering.

Activation steering modifies model behavior by intervening on hidden states during inference (Turner et al., 2023). It has been applied to honesty (Zou et al., 2023) and generation style, showing that behavioral attributes encode as linear directions in activation space. Two recent papers apply it to economic games. Ma (2026) probes and steers LLM representations in a Dictator Game, showing that injecting demographic vectors shifts giving behavior. Sun and Zhang (2026) construct persona vectors for altruism in Qwen-2.5-7B, steering at a single fixed layer across six games. They report that positive steering reliably increases prosocial behavior while negative steering has weaker effects, and flag mechanistic circuit identification as future work.

Our work differs from both in many ways. We target a Nash-specific direction benchmarked against analytically computed equilibria, probe all layers rather than fixing a single one, demonstrate that ablating the top opponent-tracking heads leaves behavior unchanged, and confirm causality through concept clamping. Our setting involves genuine repeated two-agent interaction over 50 rounds with a shared history, not one-shot games. The asymmetry Sun and Zhang (2026) observe between positive and negative steering follows from the suppression circuit we identify: positive steering amplifies an already-active cooperative mechanism while negative steering must overcome it. Gemp et al. (2024) use game-theoretic solvers to steer LLM decoding at the prompt level rather than on internal activations.

## 3 Background

Let's briefly review the game-theoretic concepts and neural network tools that our experiments rely on.

### 3.1 Nash Equilibrium

A *finite strategic game* consists of a finite set of players $N = \{1, \dots, n\}$, where each player $i$ has a finite set of actions $A_i$, a mixed strategy simplex $\Delta(A_i)$ over those actions, and a payoff function $u_i : \prod_j A_j \to \mathbb{R}$ that maps action profiles to real-valued outcomes. Players choose strategies simultaneously and independently.

A Nash equilibrium is a strategy profile $\sigma^* = (\sigma_1^*, \dots, \sigma_n^*)$ in which no player can improve their expected payoff by unilaterally deviating. Formally, for each player $i$:

$$u_i(\sigma_i^*, \sigma_{-i}^*) \geq u_i(\sigma_i, \sigma_{-i}^*) \quad \text{for all } \sigma_i \in \Delta(A_i),$$

where $\sigma_{-i}^*$ denotes the strategies of all players other than $i$. Nash (1950) proved that a Nash equilibrium exists in every finite game. Convergence from adaptive learning requires strong conditions (Daskalakis et al., 2009); no-regret learning converges only to the weaker correlated equilibrium (Hart and Mas-Colell, 2000).

### 3.2 Residual Stream and Linear Probing

A transformer with $L$ layers maintains a residual stream $\mathbf{h}_l \in \mathbb{R}^d$, a vector of dimension $d$ that is updated at each layer $l$ by attention and feed-forward sublayers. Linear probing (Alain and Bengio, 2017) trains a logistic classifier on $\mathbf{h}_l$ to test whether a concept is linearly decodable at each layer.

The logit lens applies the model's output embedding matrix $W_U$ (a learned matrix that maps the $d$-dimensional hidden state to vocabulary-size logits) directly to each intermediate hidden state $\mathbf{h}_l$. This yields a layer-wise predicted token distribution, making it possible to read off what action the model would choose if it stopped

processing at layer $l$. Concretely, throughout this paper the lens is applied at the final (decision) token position of the prompts of Appendix A: the hidden state $\mathbf{h}_l$ is passed through the final layer norm and $W_U$, and we report the probability mass on the two action-name tokens (for example `Cooperate` and `Defect`), renormalized over that pair, exactly as in the intervention readouts of Section 8.

## 4 Experimental Setup

Let us now describe the games we use, the metric we use to measure how far play is from a Nash equilibrium, and the models and protocols behind all the experiments.

### 4.1 Games and Distance Metric

We chose four two-player games that together cover the main ways Nash equilibria can be structured. Each game is small enough that the equilibria can be computed exactly, which is what lets us measure deviations precisely.

The *Prisoner's Dilemma* (PD) is the simplest test of whether a model will defect when defection is the dominant strategy. Each player chooses to Cooperate or Defect. Mutual cooperation pays (3,3), mutual defection pays (1,1), and unilateral defection pays the defector 5 while the cooperator gets 0. Defection is rational regardless of what the opponent does, so the unique Nash equilibrium is mutual defection. Yet mutual cooperation is Pareto superior, which means a model with any prosocial training will feel the pull in both directions at once.

The *Battle of the Sexes* (BoS) tests something different: not whether a model can identify the Nash equilibrium, but which one it picks. Two players must choose between Opera and Football with conflicting preferences; both pure-strategy coordinations are Nash equilibria. What matters here is which focal point each model gravitates toward, and whether two models from different families will agree.

The *Stag Hunt* (SH) has two Nash equilibria that differ in kind. Stag/Stag pays (4,4) and is payoff-dominant; Hare/Hare pays (3,3) and is risk-dominant (Harsanyi and Selten, 1988). A model that trusts its partner hunts the stag; a model that plays it safe hunts the hare alone. The game therefore tests which kind of reasoning dominates at different scales.

Finally, *Matching Pennies* (MP) is a zero-sum game with no pure-strategy equilibrium. Each player picks Heads or Tails: one player wins when they match, the other wins when they differ. The only Nash equilibrium is to randomize 50/50, which requires a kind of deliberate unpredictability that is hard to sustain over 50 rounds.

To measure how far play is from a Nash equilibrium, we track Nash distance. After $t$ rounds let $\hat{\mu}_A^{(t)}$ and $\hat{\mu}_B^{(t)}$ be the empirical mixed strategies of players A and B, and let $\mathcal{E}$ be the set of Nash equilibria for the game. Nash distance is the shortest Euclidean distance from the empirical joint strategy to any equilibrium in $\mathcal{E}$:

$$d_{\text{Nash}}^{(t)} = \min_{(\sigma_A^*, \sigma_B^*) \in \mathcal{E}} \left\| \begin{pmatrix} \hat{\mu}_A^{(t)} - \sigma_A^* \\ \hat{\mu}_B^{(t)} - \sigma_B^* \end{pmatrix} \right\|_2, \tag{1}$$

where $\sigma_A^*$ and $\sigma_B^*$ are the Nash equilibrium strategies for players A and B. A value of 0 means the pair is playing a Nash equilibrium. In Prisoner's Dilemma, a value of 2 is the maximum: it means both players have been cooperating 100% of the time, as far from mutual defection as it is possible to get. We use $d_{\text{Nash}}^{(50)}$ as the summary statistic for each experiment.

Two caveats about this metric deserve mention. First, $d_{\text{Nash}}$ measures geometric proximity of the empirical joint strategy to the nearest equilibrium; it is a distributional summary, not a measure of strategic coherence. A play sequence can be close to Nash in this metric while still being exploitable round-to-round, and a low $d_{\text{Nash}}$ should be read as "the empirical action frequencies resemble an equilibrium," not as "the agent is playing an optimal strategy." We use it because it is exactly computable against analytically derived equilibria and because it is the natural summary for the behavioral question of whether models approximate Nash frequencies. Second, because sampling is stochastic at $\tau = 0.7$, we re-ran every self-play cell across multiple

independent random seeds and report the mean and standard deviation of the Nash distance (Appendix C), which measures reproducibility directly. The seed-to-seed spread is small for the load-bearing results: each of the three larger models locks at full cooperation in Prisoner's Dilemma Direct mode ($d = 2.00$) with zero variance across all seeds, so the cooperative lock is not a single-run artifact. Several chain-of-thought and scratchpad cells are more seed-variable, and we scope the corresponding claims accordingly.

### 4.2 Models, Reasoning Conditions, and Protocol

For the behavioral and cross-play experiments we use four open-source instruction-tuned models: Llama-3-8B-Instruct, Llama-3-70B-Instruct, Qwen2.5-32B-Instruct, and Qwen2.5-72B-Instruct. Together these span 8B to 72B parameters and two distinct model families, which lets us separate scale effects from architecture effects. The 8B model runs on an NVIDIA RTX A6000 (48 GB); the larger models run on NVIDIA H200 GPUs (141 GB each), both loaded via HuggingFace Transformers in fp16 precision. The mechanistic experiments use Llama-3-8B-Instruct exclusively, loaded through TransformerLens so that we can intercept and modify activations at every one of its 32 layers. All experiments use temperature $\tau = 0.7$; Appendix D reports a deterministic-decoding (greedy, $\tau = 0$) robustness check on the Direct self-play grid, in which the headline behavioral split reproduces cell-for-cell.

Each model plays each game under three prompt structures, which we vary to understand how reasoning space affects behavior. In *Direct* mode the model is simply asked for the action name: no reasoning, just a decision. In *Chain-of-Thought* (CoT) mode it reasons step by step before committing (Wei et al., 2022), making its reasoning visible. In *Scratchpad* mode it also reasons before deciding, but the reasoning is private and not shown to the opponent. Each self-play cell runs for 50 rounds, giving 4,800 decision points per seed across the full 4 games × 3 modes × 4 models design. The cross-play experiments add all 12 ordered heterogeneous pairings across the four models, yielding a further 14,400 decision points per seed.

Table 1 summarizes the full set of experiments and where each is reported. One design detail matters for interpreting the behavioral results: the natural-language description given to the model states the equilibrium in two of the four games. The Prisoner's Dilemma description notes that mutual defection is the Nash equilibrium, and the Matching Pennies description notes the 50/50 mixed equilibrium; the Battle of the Sexes and Stag Hunt descriptions do not name an equilibrium. Appendix F tests this confound directly with a prompt-ablation ladder on the Prisoner's Dilemma structure; ablating the equilibrium statement changes no cell, while the action labels turn out to carry the behavioral and mechanistic signature. In Prisoner's Dilemma and Matching Pennies the model is therefore told the game-theoretic solution, so a model that plays the equilibrium there may be partly following the stated solution rather than deriving it. We flag this as a confound, particularly for the Prisoner's Dilemma result under the corrected pipeline, where Llama-3-8B plays close to Nash. The exact prompts, including the equilibrium-stating lines, are reproduced verbatim in Appendix A.

## 5 Behavioral Results

Before turning to the mechanistic question of why deviations occur, it is worth establishing what the deviations look like across four models and three reasoning conditions. Table 2 gives the full picture; Figure 1 shows how Nash distance evolves round by round for the 8B model.

The single most robust behavioral finding is a cooperative lock in the larger models. In Prisoner's Dilemma under Direct prompting, all three larger models (Llama-70B, Qwen-32B, and Qwen-72B) cooperate 100% of the time ($d = 2.00$, zero variance across seeds), a complete failure to play the Nash equilibrium that holds across both model families and is unaffected by the more-than-twofold spread in their sizes. Llama-8B behaves in the opposite way: without reasoning it plays close to Nash, defecting on almost every round ($d = 0.04$). So in Direct mode the failure to play Nash is a property of the larger models rather than a universal one, and Llama-8B, the model on which our mechanistic analysis is performed, is the one model that does play Nash. Section 6 shows that this single Nash-playing model makes Nash play contagious in mixed pairings.

Chain-of-thought does not uniformly break this lock, and it does not worsen the small model. The 8B already plays close to Nash in Direct mode and stays near Nash under CoT ($d = 0.04$), so reasoning does not make it

Table 1: Summary of experiments. The first two rows are behavioral; the remaining rows are mechanistic and interventionist, performed primarily on Llama-3-8B with cross-scale and cross-architecture checks on Llama-3-70B and Qwen2.5-72B.

| Experiment | Models / games | Reported in |
|---|---|---|
| Self-play behavior | all four models, all four games | Sec. 5, Table 2 |
| Cross-play behavior | 12 ordered pairings, all four games | Sec. 6, Table 3 |
| Logit lens | 8B, 70B, Qwen-72B; PD and all games | Sec. 7.1–7.4 |
| Linear probing (opponent history) | 8B; synthetic prompts | Sec. 7.1 |
| Head ablation | 8B; Prisoner's Dilemma | Sec. 7.2, Fig. 3 |
| Activation steering | 8B; PD and cross-game | Sec. 8, Fig. 8, 10 |
| Concept clamping | 8B; Prisoner's Dilemma | Sec. 8, Fig. 9 |
| Greedy-decoding robustness | all four models; all four games | App. D |
| Generation under intervention | 8B; Prisoner's Dilemma | App. E |
| Prompt-ablation ladder | all four models; PD structure | App. F |
| Base vs. RLHF | 8B base and instruct; PD | App. B |

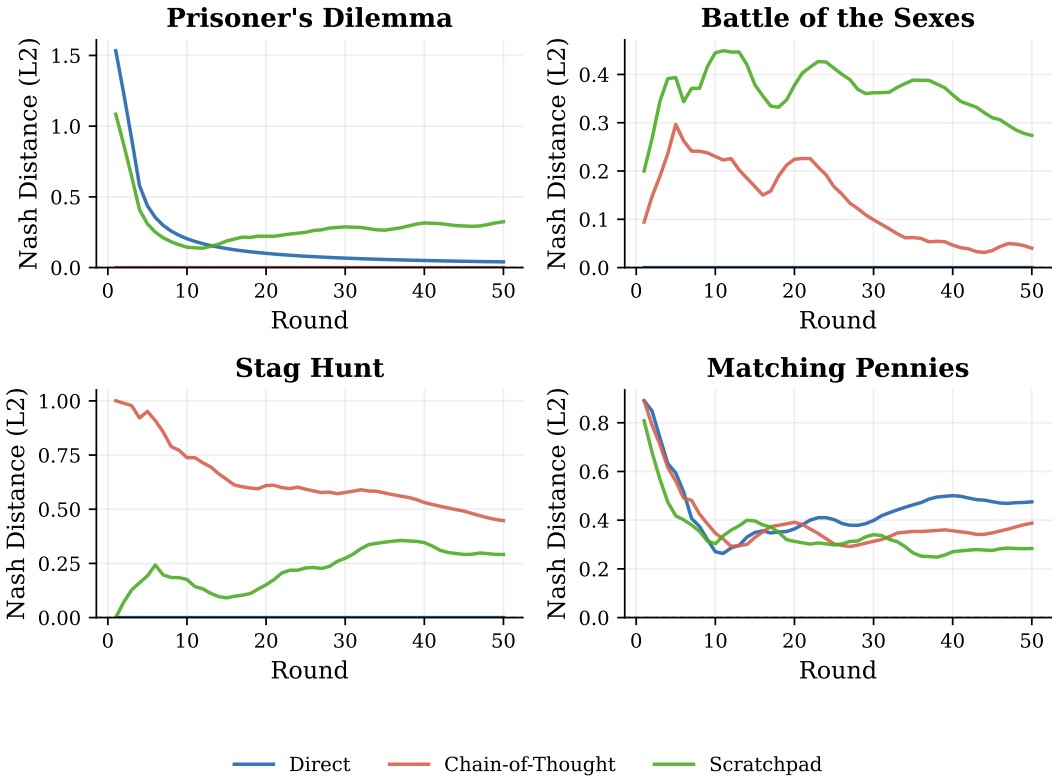

Figure 1: Nash distance over 50 rounds, Llama-3-8B self-play, single representative seed. Under Direct prompting the model plays close to Nash in several games (e.g. Prisoner's Dilemma and Stag Hunt), while chain-of-thought and scratchpad are farther from equilibrium in the coordination games and in Prisoner's Dilemma scratchpad (in Matching Pennies both are closer than Direct, and Prisoner's Dilemma CoT is comparable to Direct). Larger models differ substantially; see Table 2.

worse. Qwen-32B, which locks at full cooperation in Direct mode ($d = 2.00$), moves sharply toward Nash under CoT ($d = 0.44 \pm 0.75$; note the large seed variance), i.e. reasoning improves it on average. Llama-70B reaches near-Nash under CoT ($d = 0.00$), while Qwen-72B is variable ($d = 0.29 \pm 0.29$). The clean lock is a

Table 2: Self-play Nash distance $d^{(50)}$, mean $\pm$ SD over seeds (15 seeds for Llama-3-8B, 10 for the larger models), 50 rounds per run, under frozen code. In Prisoner's Dilemma Direct mode the three larger models fail to play Nash and lock at full cooperation ($d = 2.00$, SD $= 0$), while Llama-3-8B plays close to Nash ($d \approx 0.04$). Chain-of-thought does not worsen the 8B (it is already near Nash) and improves Qwen-32B. Several chain-of-thought and scratchpad cells show large seed variance. L-8B = Llama-3-8B, L-70B = Llama-3-70B, Q-32B = Qwen2.5-32B, Q-72B = Qwen2.5-72B.

| Game | Mode | L-8B | L-70B | Q-32B | Q-72B |
|---|---|---|---|---|---|
| Prisoner's Dilemma | Direct | $0.04 \pm 0.01$ | $2.00 \pm 0.00$ | $2.00 \pm 0.00$ | $2.00 \pm 0.00$ |
| | CoT | $0.04 \pm 0.05$ | $0.00 \pm 0.01$ | $0.44 \pm 0.75$ | $0.29 \pm 0.29$ |
| | Scratchpad | $0.43 \pm 0.10$ | $0.93 \pm 0.55$ | $0.84 \pm 0.94$ | $1.34 \pm 0.54$ |
| Battle of the Sexes | Direct | $0.01 \pm 0.02$ | $0.01 \pm 0.04$ | $0.27 \pm 0.10$ | $0.00 \pm 0.01$ |
| | CoT | $0.28 \pm 0.18$ | $0.03 \pm 0.05$ | $0.06 \pm 0.07$ | $0.02 \pm 0.03$ |
| | Scratchpad | $0.28 \pm 0.11$ | $0.10 \pm 0.09$ | $0.07 \pm 0.05$ | $0.10 \pm 0.15$ |
| Stag Hunt | Direct | $0.00 \pm 0.00$ | $0.01 \pm 0.01$ | $0.00 \pm 0.00$ | $0.00 \pm 0.00$ |
| | CoT | $0.29 \pm 0.22$ | $0.07 \pm 0.05$ | $0.05 \pm 0.04$ | $0.21 \pm 0.28$ |
| | Scratchpad | $0.54 \pm 0.12$ | $0.09 \pm 0.05$ | $0.02 \pm 0.02$ | $0.30 \pm 0.27$ |
| Matching Pennies | Direct | $0.46 \pm 0.15$ | $0.01 \pm 0.02$ | $0.00 \pm 0.01$ | $0.03 \pm 0.03$ |
| | CoT | $0.30 \pm 0.09$ | $0.22 \pm 0.14$ | $0.23 \pm 0.22$ | $0.40 \pm 0.29$ |
| | Scratchpad | $0.21 \pm 0.12$ | $0.11 \pm 0.04$ | $0.09 \pm 0.06$ | $0.14 \pm 0.09$ |

Direct-mode phenomenon in the larger models, and it holds across both model families, which is consistent with a scale effect rather than an architecture-specific one; we note, however, that we test only one sub-32B model and only one model family below 32B, so the scale reading is suggestive rather than established.

Scratchpad mode is the least reproducible condition, and the previous version's apparent architecture effect does not survive multi-seed runs. In PD Scratchpad the larger models are all highly seed-variable (Qwen-32B $0.84 \pm 0.94$, Llama-70B $0.93 \pm 0.55$, Qwen-72B $1.34 \pm 0.54$), so we can no longer claim that Qwen suppresses the cooperative prior faster than Llama; the earlier single-run contrast was not representative. We therefore scope Scratchpad claims to the observation that private reasoning induces high-variance, less predictable play rather than a clean architecture ordering.

In the two coordination games we test, Battle of the Sexes and Stag Hunt, Direct mode produces clean coordination in all four models. In Battle of the Sexes all four models reach a pure equilibrium in Direct mode ($d \leq 0.27$, and $d \leq 0.02$ for three of the four), and in Stag Hunt every model coordinates on a pure equilibrium ($d \approx 0.00$). A low Nash distance in these games therefore reflects successful coordination on one of the equilibria rather than Nash avoidance. Under chain-of-thought and scratchpad the coordination-game cells become seed-variable (for example Llama-8B Stag Hunt CoT $0.29 \pm 0.22$, Scratchpad $0.54 \pm 0.12$), and we do not read specific equilibrium selections (payoff-dominant versus risk-dominant) off single-run values as the previous version did.

Matching Pennies is the hardest game for these models. In Direct mode the 8B is the outlier ($d = 0.46$), tending toward a pure action rather than the required 50/50 mix, while the larger models stay near the mixed target ($d \leq 0.03$). The chain-of-thought Matching Pennies cells are high-variance across models (Qwen-72B $0.40 \pm 0.29$, Qwen-32B $0.23 \pm 0.22$, Llama-70B $0.22 \pm 0.14$), so we report them as unstable rather than singling out any one cell as extremal.

Whether these deviations can be controlled is the question the mechanistic analysis addresses.

# 6 Cross-Play and Heterogeneous Agent Interactions

Self-play tells us how each model behaves against a copy of itself, but real deployments involve agents that differ. To see what happens when models from different families and scales meet each other, we ran all 12

Table 3: Cross-play Nash distance $d^{(50)}$ in *Direct* mode, mean $\pm$ SD over 10 seeds, for all 12 ordered heterogeneous pairings (Agent A vs Agent B). A single 8B model in the pairing collapses Prisoner's Dilemma to mutual defection ($d \approx 0.08$); every pairing of two larger models sustains the cooperative lock ($d = 2.00$). Coordination games (BoS, SH) coordinate almost everywhere; Matching Pennies stays near its mixed target for the two-large-model pairings but deviates in the pairings containing the 8B ($d \approx 0.3$). The full 144-cell table across all three reasoning modes is available online (link omitted for anonymous review). L-8B = Llama-3-8B, L-70B = Llama-3-70B, Q-32B = Qwen2.5-32B, Q-72B = Qwen2.5-72B.

| Agent A | Agent B | PD | BoS | SH | MP |
|---------|---------|-----|-----|-----|-----|
| *Mixed pairings (one 8B model present)* | | | | | |
| L-70B | L-8B | $0.07 \pm 0.02$ | $0.01 \pm 0.02$ | $0.00 \pm 0.00$ | $0.34 \pm 0.16$ |
| L-8B | L-70B | $0.07 \pm 0.02$ | $0.00 \pm 0.01$ | $0.07 \pm 0.07$ | $0.30 \pm 0.12$ |
| L-8B | Q-32B | $0.09 \pm 0.04$ | $0.03 \pm 0.00$ | $0.00 \pm 0.00$ | $0.32 \pm 0.16$ |
| L-8B | Q-72B | $0.07 \pm 0.02$ | $0.01 \pm 0.01$ | $0.00 \pm 0.00$ | $0.34 \pm 0.13$ |
| Q-32B | L-8B | $0.09 \pm 0.04$ | $0.01 \pm 0.01$ | $0.00 \pm 0.00$ | $0.34 \pm 0.17$ |
| Q-72B | L-8B | $0.07 \pm 0.02$ | $0.01 \pm 0.01$ | $0.00 \pm 0.00$ | $0.26 \pm 0.13$ |
| *Both models $\geq$ 32B* | | | | | |
| L-70B | Q-32B | $2.00 \pm 0.00$ | $0.11 \pm 0.10$ | $0.00 \pm 0.00$ | $0.01 \pm 0.02$ |
| L-70B | Q-72B | $2.00 \pm 0.00$ | $0.01 \pm 0.02$ | $0.00 \pm 0.00$ | $0.02 \pm 0.02$ |
| Q-32B | L-70B | $2.00 \pm 0.00$ | $0.00 \pm 0.00$ | $0.01 \pm 0.01$ | $0.01 \pm 0.01$ |
| Q-32B | Q-72B | $2.00 \pm 0.00$ | $0.01 \pm 0.01$ | $0.00 \pm 0.00$ | $0.01 \pm 0.02$ |
| Q-72B | L-70B | $2.00 \pm 0.00$ | $0.00 \pm 0.00$ | $0.01 \pm 0.01$ | $0.03 \pm 0.03$ |
| Q-72B | Q-32B | $2.00 \pm 0.00$ | $0.12 \pm 0.09$ | $0.00 \pm 0.00$ | $0.02 \pm 0.02$ |

ordered pairings across our four models in all three reasoning modes and all four games, 50 rounds each, yielding 144 experimental cells and 14,400 decision points per seed. What we found cannot be predicted from self-play alone.

The first thing that stands out is what we call the 8B defection unlock. Under the corrected pipeline Llama-8B itself plays close to Nash in Prisoner's Dilemma Direct mode, defecting almost every round (self-play $d = 0.04$), and its presence in a pairing makes Nash play contagious: every mixed pairing that contains the 8B collapses to mutual defection ($d \approx 0.08$, SD $\approx 0.02$; Table 3), even though the larger models cooperate 100% in self-play and in every all-large pairing. The effect is symmetric in whether the 8B is Agent A or Agent B (for example, L-70B vs L-8B and L-8B vs L-70B are both 0.07), so it is not an ordering artifact: the 8B defects, the larger partner best-responds to a defector by defecting, and mutual defection follows without any strategic reasoning. The opposite holds when both models are large. In PD Direct, every pairing of two models $\geq$ 32B produces $d = 2.00$ with zero variance: both cooperate 100% indefinitely, each sustaining the other's cooperative prior. Chain-of-thought weakens both effects but does not cleanly dissolve them. With CoT, most cross-play pairings move toward Nash in PD, but with substantial seed variance, and the all-large pairings in particular remain variable (mean $d$ ranging from about 0.01 to 0.61 across pairings). So reasoning reduces the cooperative prior in mixed pairings but does not reliably eliminate it between two large models. Beyond PD, the cross-play picture is dominated by variance rather than clean secondary effects, and we scope it accordingly. In Stag Hunt, Direct-mode pairings coordinate on a pure equilibrium almost everywhere ($d \approx 0.00$), while the chain-of-thought and scratchpad Stag Hunt cells are seed-variable; we do not attribute specific equilibrium selections (payoff-dominant versus risk-dominant) to particular pairings, because the single-run values that suggested them do not reproduce. Battle of the Sexes coordinates in Direct mode across pairings ($d$ at or below 0.12). Matching Pennies remains the hardest game: mixed pairings containing the 8B stay away from the mixed equilibrium under chain-of-thought (for example L-8B vs Q-72B $d = 0.67$, Q-72B vs L-8B $d = 0.70$, Q-32B vs L-8B $d = 0.49$), consistent with the 8B's difficulty randomizing, but these cells are high-variance and we do not rank them as extremal. A common failure in these cells is that Llama-8B with CoT locks onto a pure strategy, the opponent responds with the opposite pure strategy, and the pair settles away from the mixed equilibrium that Matching Pennies requires. Direct mode is closer to the

mixed target for Matching Pennies across pairings. Finally, we note that the corrected data does not support a robust role-order effect. In Direct mode, the clean condition, the outcome is symmetric in whether a given model is Agent A or Agent B (for example L-70B vs L-8B and L-8B vs L-70B are both $d = 0.07$, and the all-large pairings are 2.00 in either order). Apparent order effects appear only in the chain-of-thought and scratchpad cells, which are highly seed-variable (for example the two orderings of Qwen-72B and Qwen-32B in PD Scratchpad differ by less than their standard deviations), so we do not treat role order as a structural effect; it is within the run-to-run variance of those conditions.

## 7 Mechanistic Analysis

All mechanistic experiments in this section use Llama-3-8B-Instruct, a 32-layer transformer loaded through TransformerLens for layer-by-layer activation access. Layer numbers (0 through 31) refer to this model throughout. The probing, logit-lens, and steering analyses all use the same synthetic prompt format (system prompt plus game description and constructed history, as a plain string) so that the three analyses are directly comparable. We note that this format does not apply the model's chat template, whereas the behavioral experiments do, and that the chat template is the variable whose application changed the 8B's Prisoner's Dilemma behavior. We therefore treat the mechanistic results as characterizing the model's internal computation on a fixed, interpretable prompt format, and read the "consistent with behavior" statements below as directional rather than as exact reproductions of the behavioral generation setup. Sections 7.3 and 7.4 extend the logit lens analysis to Llama-3-70B-Instruct and Qwen2.5-72B-Instruct respectively, to test whether the circuit structure holds at larger scale and across architectures.

### 7.1 Opponent History and the Nash Preference

We report one clean, single-provenance probing result and otherwise let the logit lens carry the mechanistic analysis. We construct 300 synthetic prompts as contrastive pairs, built exactly as for the steering extraction (Section 8); 150 have a history in which the opponent plays the Nash action and 150 in which it plays the opposite, so that the probing and steering analyses share one prompt format and one data source. On the hidden state $\mathbf{h}_l$ at the final decision token we train a logistic probe for the opponent's last action (labels balanced by construction, 5-fold cross-validation, chance = 0.5). The opponent's last action is perfectly linearly decodable: the probe reaches 1.00 at every layer. We note that on these contrastive prompts the two conditions (opponent-plays-Nash vs opponent-plays-opposite) use homogeneous histories, so they are perfectly linearly separable at every layer; this establishes that the opponent's action is present in the residual stream, but does not on its own establish fine-grained opponent tracking across mixed histories, which we do not test here.

We deliberately do not report a Nash-action decodability number. Defining the model's action on a prompt as the argmax over the two action tokens at the decision position, the 8B's action is Defect (the Nash action) on all 300 of these constructed Prisoner's Dilemma prompts, so the Nash-action label has no variance and a probe for it is undefined. (This concerns the argmax label, not sampled play: the mean probability mass on Defect on these prompts is 0.616, reported in Section 8, so the model favors Defect on every prompt without placing all its mass there. This is itself consistent with the behavioral finding that the 8B plays Nash, and it corrects a claim in an earlier version of this paper: an apparent "Nash decodability" signal there was an artifact of pooling trajectories from two code versions, which a probe can separate by prompt format; we do not rely on any such number.) What a probe cannot show here, the logit lens can.

The logit lens reveals the internal competition. Layers 0 through 23 assign majority probability to Defect, the Nash action in Prisoner's Dilemma. At layer 24 the picture inverts: $P(\text{Cooperate})$ surges and rises to a peak of about 0.80 at layer 30, before the final layer commits back to Defect ($P(\text{Defect}) \approx 0.68$, i.e. $P(\text{Cooperate}) \approx 0.32$). This late-layer cooperative surge (Figure 2) is the computational signature of the override. Across games, the same qualitative pattern holds: through the middle of the network the logit lens favors the game's focal action (its pure Nash action where one exists; Heads in Matching Pennies), and it then flips away from that action late, with the onset between layers 24 and 29 (Table 4); the per-game focal convention and the Matching Pennies case are defined in the caption of Table 4.

Table 4: Layer at which the logit lens first flips away from the model's mid-network preferred action (the override onset) for Llama-3-8B, by game. Logit-lens-derived (not probe-derived). For games with more than one pure equilibrium, the focal convention is Opera in Battle of the Sexes and Stag in Stag Hunt; for Matching Pennies, which has no pure equilibrium, the onset marks the layer at which the mid-network preferred action (Heads) flips to the other (Tails).

| Game | Override-onset layer |
|---|---|
| Prisoner's Dilemma | 24 |
| Battle of the Sexes | 29 |
| Stag Hunt | 28 |
| Matching Pennies | 26 |

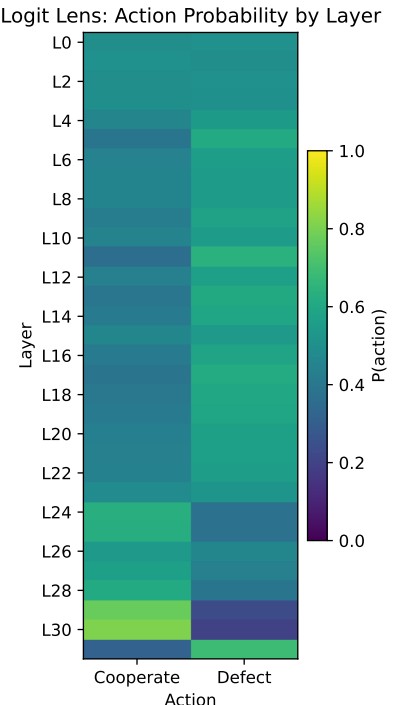

Figure 2: Logit lens in Prisoner's Dilemma, averaged over 20 synthetic contrastive prompts of increasing history length (the same construction as Section 7.1; the comparison panels use the same prompts). Layers 0–23 favor Defect (the Nash action). At layer 24, Cooperate surges and peaks at $P(\text{Cooperate}) \approx 0.80$ at layer 30, before layer 31 commits to Defect.

## 7.2 Where the Override Lives and Head Ablation

Having identified the late-layer override through the logit lens, we ask whether it can be attributed to specific attention heads. We score each head by the weight it places on tokens corresponding to the opponent's past actions and zero-ablate the top-scoring ones. Figure 3 shows the result. Ablating each of the top-5 opponent-tracking heads individually and all five jointly produces $\Delta P(\text{Nash}) = 0.000$ in every condition, where $\Delta P(\text{Nash})$ denotes the change in probability of the Nash action relative to the unablated baseline. The cooperative override is a distributed residual-stream effect; no head or combination of the top opponent-tracking heads we ablate carries the cooperative bias (we ablate heads selected for opponent-tracking, so this bounds those heads specifically rather than all heads). The logit lens separately shows this override building gradually across the late layers rather than appearing at a single point. This is why activation steering, which operates on the full residual stream vector, succeeds where head ablation fails.

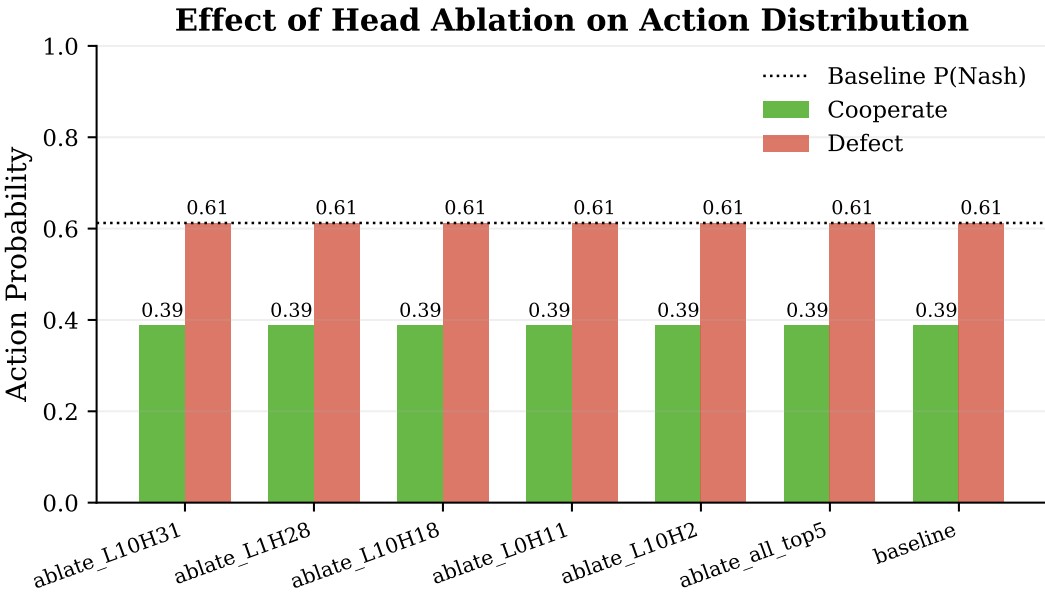

Figure 3: Zero-ablating the top-5 opponent-tracking heads individually and jointly produces no change in action distribution ($\Delta P$(Nash) = 0.000 in every case). Green bars indicate Cooperate probabilities; red bars indicate Defect. The cooperative override is not carried by the top opponent-tracking heads we ablate (this bounds those heads, not all heads).

### 7.3 Cross-Scale Validation on Llama-3-70B-Instruct

To test whether the circuit structure identified in the 8B model holds at larger scale, we ran the logit lens analysis on Llama-3-70B-Instruct (80 layers) using HuggingFace with `device_map=auto` distributed across eight NVIDIA H200 GPUs (141 GB each) on the same shared storage infrastructure as the behavioral experiments. The 70B model uses the same Llama-3 architecture as the 8B model, making the comparison clean.

Figure 4 shows the side-by-side logit lens comparison. Two findings stand out.

First, the cooperative override is present in both models, and in both the final layer resolves it toward the Nash action. In the 8B model, $P$(Cooperate) peaks at approximately 0.80 at layer 30, and the final layer then commits to Defect ($P$(Cooperate) $\approx$ 0.32), consistent with the 8B playing near-Nash behaviorally. In the 70B model, a cooperative bump appears in layers 60–75 peaking at approximately 0.68, and the final-layer logit lens settles at $P$(Cooperate) = 0.45. These logit-lens probabilities are computed on the synthetic prompts, which are not tokenized identically to the behavioral generation setup (no chat template is applied, and the histories are synthetic); they indicate the direction of the internal computation, not the exact behavioral action frequencies, which is why the 70B's final-layer value sits below 0.5 even though the 70B cooperates in its actual gameplay, and why the 8B's 0.32 is directional rather than equal to its $\sim$2–3% behavioral cooperation. The override thus exists at both scales and is overridden in turn at the final layer of each; we do not claim a monotonic scale trend in the strength of this correction, since the 8B's final-layer readout is at least as Nash-committed as the 70B's. What differs behaviorally is that the larger models sustain full cooperation in self-play Direct mode while the 8B does not, and the mechanistic picture we can state with confidence is that the override is a general feature of the residual stream across scale, not that a single layer-wise number explains the behavioral difference.

Second, the cross-game logit lens for 70B (Figure 5) shows the final layers strongly favoring the Nash action in Battle of the Sexes and Stag Hunt, with $P$(non-Nash) collapsing to near zero in the final layers. Matching Pennies stays near chance throughout. This explains behaviorally why 70B achieves near-Nash play across most games even in Direct mode, while remaining locked in cooperation for PD.

Figure 4: Logit lens comparison: Llama-3-8B (left) vs Llama-3-70B (right), Prisoner's Dilemma. The 8B model shows the cooperative override peaking at approximately 0.80 with the final layer committing to Defect ($P(\text{Cooperate}) \approx 0.32$). The 70B model shows a weaker cooperative bump (peak 0.68) with the final layer settling at $P(\text{Cooperate}) = 0.45$. The override exists at both scales; the final layer of each resolves toward the Nash action.

Taken together, the 70B results support the main mechanistic claim: the cooperative override is a general property of the Llama model family, not an artifact of the 8B model's size. What the 70B adds is that the override, and a late-layer correction toward Nash, are present at larger scale as well.

### 7.4 Qwen2.5-72B: A Different Suppression Architecture

To test whether the suppression mechanism generalizes beyond the Llama model family, we ran the same logit lens analysis on Qwen2.5-72B-Instruct (80 layers), a model from a different architecture family that also shows strong cooperative behavior in the behavioral experiments.

The results reveal a mechanistically distinct pattern. Figure 6 compares the logit lens with the 8B baseline. In Qwen, layers 0–20 hover near chance, layers 20–35 show a modest cooperative bump, and then layers 35–70 show a dramatic collapse toward Nash: $P(\text{Cooperate})$ drops near zero, meaning the model is strongly computing Defect through most of its mid-to-late network. Then, in the final layers 70–80, there is a sharp reversal: $P(\text{Cooperate})$ spikes to 0.99. The cooperative override in Qwen is not a gradual late-layer buildup as in Llama-3-8B: it is an abrupt final-layer reversal that overrides a strongly Nash-favoring mid-network.

The cross-game logit lens (Figure 7) confirms this pattern: mid-network the model favors each game's focal action (its pure Nash action where one exists; in Matching Pennies, near-pure Heads, which is in fact far from the 50/50 equilibrium), followed by a sharp late-network spike away from that mid-network action, toward the cooperative action in Prisoner's Dilemma and toward a different equilibrium action in the coordination games (Section 8.3).

These results establish two conclusions. First, the cooperative override is not a Llama-specific artifact: it is present in Qwen too, suggesting it is a general property of large instruction-tuned models. Second, the two architectures implement the override differently. Llama-70B and Qwen-72B both cooperate behaviorally in Direct mode, but internally the Llama-family lens shows a gradual late-layer buildup of the cooperative signal (with a partial final-layer correction), whereas Qwen computes the Nash action strongly through most of its depth and then reverses abruptly in the final layers. We do not claim that either architecture's private reasoning suppresses the cooperative prior more effectively than the other, as the scratchpad cells are high-variance and do not support such an ordering.

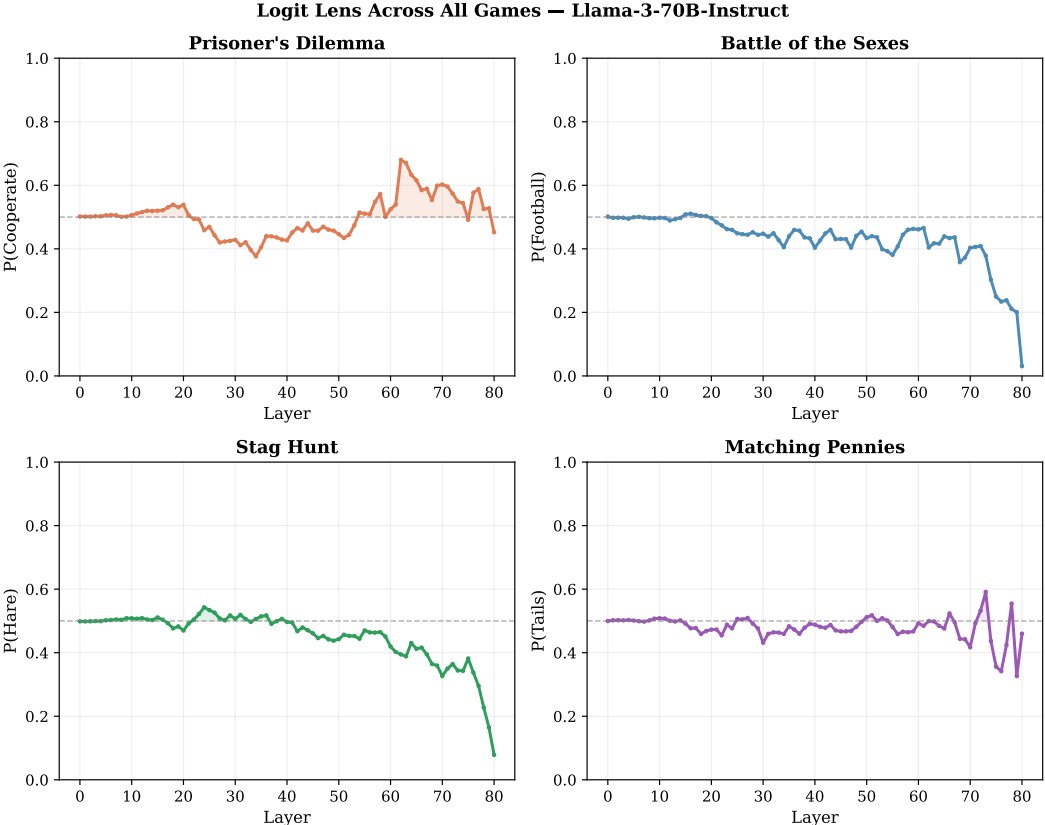

Figure 5: Cross-game logit lens for Llama-3-70B (80 layers). Prisoner's Dilemma shows the cooperative bump in layers 60–75. Throughout, "the Nash action" in a game with more than one pure equilibrium denotes the model's baseline focal equilibrium action (Opera in Battle of the Sexes, Stag in Stag Hunt); Matching Pennies has no pure equilibrium. Battle of the Sexes and Stag Hunt show strong final-layer collapse toward the Nash action. Matching Pennies stays near chance throughout, with modest late-layer oscillation.

## 8 Intervention Experiments

The logit lens results tell us where the cooperative bias lives; the following experiments ask whether it can be moved.

### 8.1 Extracting and Steering the Nash Direction

The head ablation result tells us the cooperative override is not in any single head, which raises the question of where exactly it is. The answer from the logit lens work, together with the head ablation, is that it lives in the residual stream as a linear direction. We extract this direction by constructing contrastive prompt pairs: prompts whose history is all mutual cooperation (both players Cooperate every prior round) versus prompts whose history is all mutual defection (both players Defect every prior round); the two conditions thus differ in both players' past actions, holding the prompt structure fixed. The cooperative direction $\mathbf{v}_{\text{coop}}$ is the mean difference in hidden states at layer $l^* = 22$, chosen just upstream of the override onset that the logit lens locates near layer 24, so that the injected direction can influence the override as it forms:

$$\mathbf{v}_{\text{coop}} = \mathbb{E}_{\text{coop}}[\mathbf{h}_{l^*}] - \mathbb{E}_{\text{defect}}[\mathbf{h}_{l^*}], \quad \|\mathbf{v}_{\text{coop}}\|_2 = 1. \tag{2}$$

As a sanity check, the top principal component of the pooled hidden states aligns with $\mathbf{v}_{\text{coop}}$, an independent decomposition recovering the same direction, which gives us confidence we are looking at something real. (We do not use the normal vector of a trained probe here: on this contrastive set the classes are perfectly linearly

Figure 6: Logit lens comparison: Llama-3-8B (left) vs Qwen2.5-72B (right), Prisoner's Dilemma. Qwen shows a dramatically different pattern: strong Nash-favoring through layers 35–70 followed by an abrupt cooperative spike ($P(\text{Cooperate}) = 0.99$) in the final layers. The cooperative override architecture differs qualitatively between model families.

separable, so a logistic decision boundary's normal is regularization-dependent and not a stable direction, which is precisely why we rely on the mean-difference and PCA directions instead.)

A natural concern is whether $\mathbf{v}_{\text{coop}}$ is a Nash-specific direction or simply a Cooperate/Defect action direction in the Prisoner's Dilemma context. In PD, these are equivalent by definition: Nash = Defect, so a direction that steers toward Nash is necessarily a direction that steers toward Defect. The cross-game steering results below (Section 8.3, Figure 10) argue against a PD-specific interpretation: the *same* PD-extracted direction produces graded, directionally consistent control in Battle of the Sexes and Stag Hunt as well, so the direction is not tied to the "Defect" token. What that shift targets in each game is not uniformly the cooperative action, however, and we return to this point when we steer across games (Section 8.3).

We then ask what happens when we push this direction up or down. Injecting $\alpha \, \mathbf{v}_{\text{coop}}$ at layers 20, 22, and 24 and sweeping $\alpha$ from $-20$ to $+40$, the effect is large and graded (Figure 8); the response is directionally consistent but not strictly monotonic: there is a local $P(\text{Defect})$ bump near $\alpha = +5$ before cooperation takes over at larger positive $\alpha$. At baseline ($\alpha = 0$) the mean probability mass on Defect is about 62% (consistent with the argmax action being Defect on these prompts; Section 7.1). Pull the cooperative direction down to $\alpha = -5$ and defection jumps to 99.2%: the model is essentially a Nash player. Push it up to $\alpha = +10$ and cooperation reaches 88.7%: the cooperative prior completely wins. The model's strategic behavior turns out to be a simple dial. (The baseline here, $P(\text{Defect}) = 0.616$, is measured on the contrastive prompts used to extract $\mathbf{v}_{\text{coop}}$; the cross-game steering experiment below, Figure 10, uses a different per-game prompt set and so has a different Prisoner's Dilemma baseline: indeed the baseline argmax action differs between the two synthetic sets (Defect on the contrastive set here, Cooperate on the per-game set of Section 8.3), so the two synthetic constructions straddle the 50% line and neither matches the $\sim$97–98% behavioral defection rate. The two experiments are therefore not directly comparable at $\alpha = 0$, only in the direction and magnitude of the response to $\alpha$.) Throughout this section, reported action probabilities are renormalized over the two action tokens (Cooperate, Defect) at the decision position; Appendix E documents free-running generation under the same interventions.

## 8.2 Confirming Causality through Concept Clamping

A reasonable worry is that steering works for reasons unrelated to the cooperative direction itself. Perhaps we are simply injecting noise, or shifting the residual stream in a way that happens to trigger cooperation through some other mechanism. Concept clamping rules this out. Rather than adding a multiple of $\mathbf{v}_{\text{coop}}$, we remove the existing cooperative component and replace it with a fixed scalar $c$:

$$\mathbf{h}_{l*} \leftarrow \mathbf{h}_{l*} - (\mathbf{h}_{l*} \cdot \hat{\mathbf{v}}) \, \hat{\mathbf{v}} + c \, \hat{\mathbf{v}}, \tag{3}$$

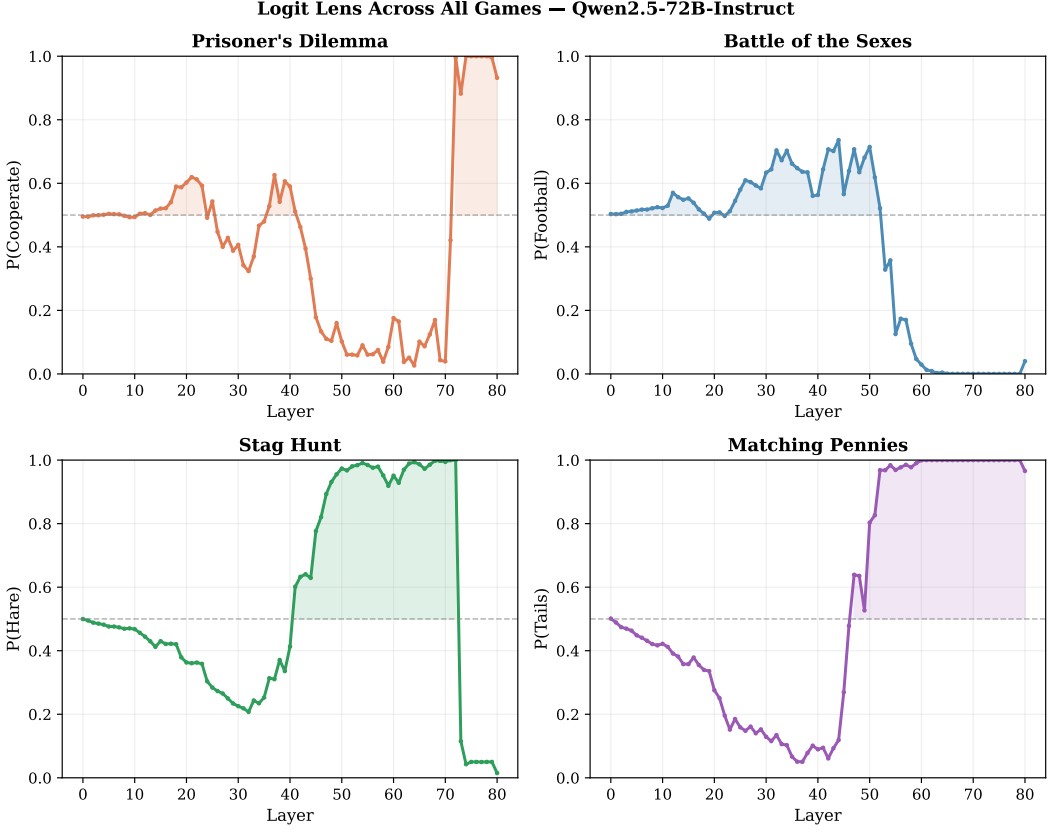

Figure 7: Cross-game logit lens for Qwen2.5-72B. Each panel plots one action probability per game (P(Cooperate) in Prisoner's Dilemma, P(Football) in Battle of the Sexes, P(Hare) in Stag Hunt, P(Tails) in Matching Pennies). Mid-network the model favors Defect, Football, Hare, and Heads respectively (so the plotted curve sits low in Prisoner's Dilemma and Matching Pennies and high in Battle of the Sexes and Stag Hunt) before the late layers move sharply away. The override is architecture-wide, not game-specific.

where $\hat{\mathbf{v}}$ is the unit cooperative direction (Eq. 2). Now $c$ is the only thing that changes across trials. If the direction is genuinely causal, $P(\text{Cooperate})$ should increase with $c$. It does, directionally and across the full range (Figure 9): sweeping $c \in [-30, 30]$ over 20 values, $P(\text{Cooperate})$ rises from 0.1% at $c = -30$ to 98.6% at $c = +30$ (Spearman $\rho = 0.72$ between $c$ and $P(\text{Cooperate})$). As in the steering sweep, the response is not strictly monotonic point-to-point, with a local defection bump near $c \approx +5$. The cooperative direction is not a bystander; it is the mechanism.

## 8.3 Does the Direction Generalize Across Games?

The steering and clamping experiments above use Prisoner's Dilemma, where the Nash action is Defect, so a reasonable objection is that $\mathbf{v}_{\text{coop}}$ might encode a PD-specific "Defect" direction rather than a direction with broader reach. To test this, we take the *same* $\mathbf{v}_{\text{coop}}$ extracted from Prisoner's Dilemma and apply it, without re-extraction and at the same injection layers (20, 22, 24), to the other three games.

Figure 10 shows the result. The direction exerts graded, directionally consistent control over action selection in three of the four games (not strictly monotonic at every step, but moving cleanly from one equilibrium action to the other as $\alpha$ increases). In Prisoner's Dilemma, negative $\alpha$ drives the model to the Nash action ($P(\text{Defect}) = 0.99$ at $\alpha = -5$) and positive $\alpha$ to Cooperate (0.98 at $\alpha = +5$). In Battle of the Sexes, positive $\alpha$ moves play from the Opera equilibrium at baseline ($P(\text{Opera}) = 0.98$) to the Football equilibrium ($P(\text{Football}) = 0.94$ at $\alpha = +20$). In Stag Hunt, sweeping $\alpha$ moves play between the two equilibria, from Stag at negative $\alpha$ ($P(\text{Stag}) = 1.00$ at $\alpha = -5$) to Hare at positive $\alpha$ ($P(\text{Hare}) = 0.92$ at $\alpha = +20$). A single

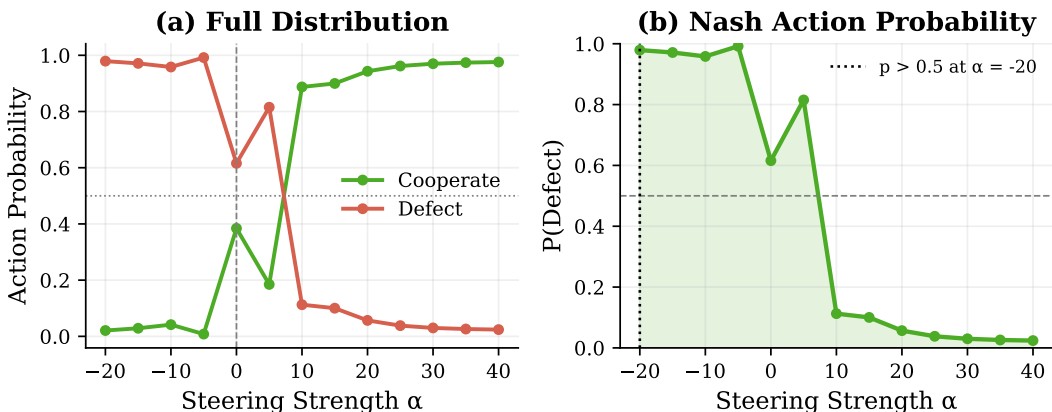

Figure 8: Steering sweep across $\alpha \in [-20, 40]$. Baseline ($\alpha = 0$): $P(\text{Nash/Defect}) = 0.616$. At $\alpha = -5$: $P(\text{Nash/Defect}) = 0.992$. At $\alpha = +10$: $P(\text{Cooperate}) = 0.887$. In panel (b), $P(\text{Defect})$ (the Nash-action probability) stays above 0.5 across the negative-$\alpha$ range and up to roughly $\alpha \approx +7$, falling below it only under strong positive steering; the "$p$" in the panel is this probability, not a significance value.

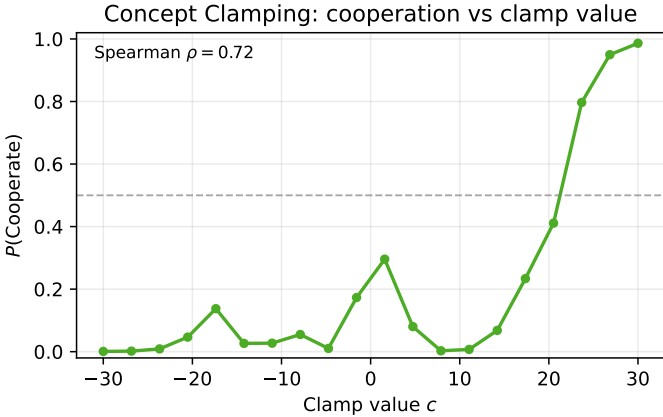

Figure 9: Concept clamping in Prisoner's Dilemma. Replacing the cooperative component of the residual stream with a fixed scalar $c$ (Eq. 3) and sweeping $c$ over 20 values, $P(\text{Cooperate})$ moves from near 0 to near 1 with strong rank correlation (Spearman $\rho = 0.72$), though not strictly monotonically point-to-point. The cooperative direction causally controls the action, rather than merely correlating with it.

direction extracted from one game thus produces systematic, graded behavioral change in three games with different action labels and equilibrium structure, which a PD-"Defect" token feature could not do.

We are careful about what this does and does not show. The target of the shift is the cooperative action only in Prisoner's Dilemma. In the two coordination games the direction selects between Nash equilibria rather than moving toward a prosocial action; in Stag Hunt in particular, positive $\alpha$ drives play toward Hare, the risk-dominant equilibrium, which is the *less* cooperative of the two (Stag, the payoff-dominant equilibrium, is the cooperative choice). In zero-sum Matching Pennies, which has no cooperative action, the response is non-monotonic and unstable: positive $\alpha$ first raises $P(\text{Tails})$ but reverts to near-pure Heads by $\alpha = +20$, so we exclude Matching Pennies from this analysis. The correct reading is therefore narrower than a cooperative prior: $\mathbf{v}_{\text{coop}}$ is a general Nash-versus-alternative control direction, not a PD-specific Defect feature but also not a uniformly cooperative one. It coincides with cooperation in Prisoner's Dilemma and with equilibrium selection in the coordination games.

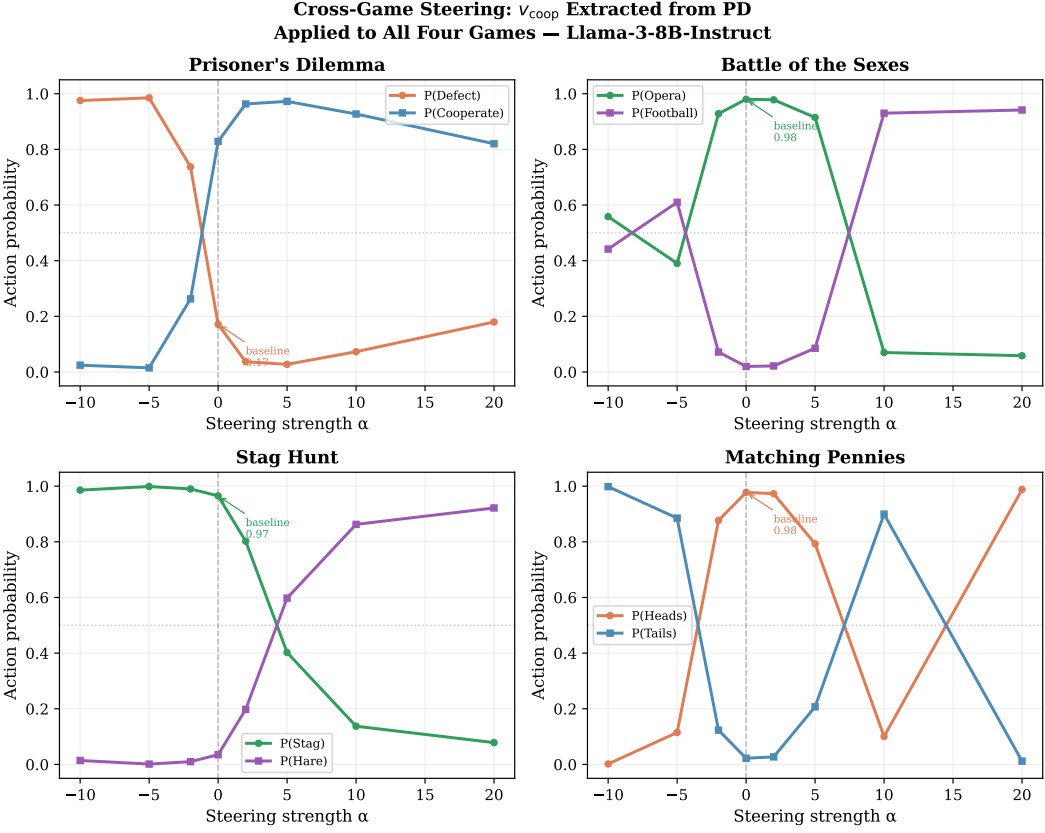

Figure 10: Cross-game steering with $\mathbf{v}_{\text{coop}}$ extracted from Prisoner's Dilemma and applied unchanged, at the same injection layers, to all four games. Sweeping $\alpha$ gives graded, directionally consistent control (not strictly monotonic at every step) in Prisoner's Dilemma (Defect $\leftrightarrow$ Cooperate), Battle of the Sexes (Opera $\leftrightarrow$ Football), and Stag Hunt (Stag $\leftrightarrow$ Hare); Matching Pennies (zero-sum) is unstable and is excluded. A single PD-derived direction thus controls action and equilibrium selection across games, which rules out a PD-specific Defect feature. The target is the cooperative action in Prisoner's Dilemma but the risk-dominant equilibrium (Hare) in Stag Hunt, so the direction is not a uniform cooperative prior.

## 9 Discussion

We now pull together the mechanistic and behavioral results, place them in the context of prior interpretability work, and draw out what they mean in practice.

### 9.1 What Generates the Cooperative Override, and When It Prevails

A clean linear probe decodes the opponent's action from the residual stream perfectly at every layer on the contrastive prompts (Section 7.1), so the opponent's action is present in the residual stream throughout. The Nash preference itself is visible not through a probe (the 8B's argmax action on the synthetic prompts is Nash on every one, so a Nash-action probe has no label variance) but through the logit lens, and the cooperative override that competes with it is not localized to the top opponent-tracking heads we ablate (Section 7.2). Through roughly the first three-quarters of the network the intermediate predictions actually lean toward defection, the Nash action. Then something flips. In the final quarter, a distributed cooperative override takes hold, peaking at about an 80% probability of cooperation before the last layer commits back toward the Nash action. The override is distributed enough that ablating the top opponent-tracking heads does not change the action distribution, but it is also concentrated enough in a geometric direction that a small push at the mid-to-late layers (20, 22, and 24) shifts the outcome from 62% to 99% defection.

The behavioral results across four models fit this picture well. Without reasoning, the three larger models default to full cooperation in the Prisoner's Dilemma (they cooperate 100% of the time), while the 8B plays close to Nash. The cooperative prior that the larger models exhibit is not a quirk of any particular model family; the base-model comparison indicates it is shaped by pretraining on human-generated text, with RLHF partially moderating but not creating the bias (see Appendix B). Chain-of-thought does not uniformly break through it, and it does not worsen the small model. The 8B already plays close to Nash in Direct mode and stays near Nash under CoT, so reasoning does not make it worse; Qwen-32B, which locks at full cooperation in Direct mode, moves sharply toward Nash under CoT; Llama-70B reaches near-Nash under CoT while Qwen-72B remains variable. We do not single out any one reasoning cell as an edge case, since the reasoning-mode cells are high-variance across models (Table 2).

Prior mechanistic work has identified circuits that are localized, either to specific heads (McDougall et al., 2023) or to specific MLP layers (Meng et al., 2022). The cooperative override appears to be neither: it is not carried by the top opponent-tracking heads we ablate, and the logit lens shows it building gradually across layers rather than at a single point (we did not run an MLP-layer ablation). The closest thing in the literature is the late-layer representational shift described by Belrose et al. (2023), but that work treats the shift as an observation. Here we show it changes with game history and can be moved by steering, which makes it a causal mechanism rather than a description. The asymmetry Sun and Zhang (2026) find between positive and negative steering also makes sense now: pushing the cooperative direction up amplifies an already-active mechanism, while pushing it down requires overcoming one.

The question is no longer why LLMs fail to play Nash. It is what generates the late-layer cooperative override, when that override prevails behaviorally (in the larger models) and when it is itself overridden (in the 8B), and whether it can be controlled. The mechanistic experiments on Llama-3-70B and Qwen2.5-72B (Sections 7.3 and 7.4) show the override is present across both scales and both model families, but is implemented differently: Llama builds it gradually through the final third of the network, while Qwen computes Nash strongly through most of its depth and reverses it abruptly in the final layers. The answer to whether it can be controlled is yes.

### 9.2 Deployment Implications and Limitations

One scope boundary sharpened by the prompt-ablation ladder (Appendix F) should be stated first. The behavioral and mechanistic findings of this paper are properties of the canonical games as presented, and they are keyed to the prompt surface in specific, now-measured ways: the cooperative lock of the larger models and the 8B's late-layer override signature are engaged by the action labels Cooperate and Defect (both vanish under neutral labels on an identical payoff structure), and the 8B's near-Nash play is engaged by the game framing (it cooperates on the identical matrix presented as a neutral decision task). Ablating the explicit

equilibrium statement, by contrast, changes nothing. We therefore do not claim that any model computes equilibria from payoff structure alone; the claims are about how these models play the canonical games, and Appendix F maps which surface features carry the effects.

The practical upshot for anyone deploying LLM agents in strategic settings is that a cooperative prior is strong in the larger models, which fail to play Nash and lock into cooperation, though it is not universal: the 8B plays close to Nash. Mechanistically, on contrastive prompts the opponent's action is perfectly separable in the residual stream at every layer (a near-trivial ceiling on those homogeneous histories, not a claim of fine-grained tracking). The logit lens favors the Nash action (Defect) through most of the network, and the cooperative override appears late and surges in the late layers, but whether it prevails depends on the model: in the 8B the final layer overrides it and the model plays Nash, whereas the larger models sustain cooperation. The good news is that this can be shifted at inference time without retraining: a small intervention at the mid-to-late layers (20, 22, and 24) moves the model between defection-dominant and cooperation-dominant play (from a baseline at which the mean probability mass on Defect is about 62%).

The cross-play results add a dimension that does not show up in any single-model evaluation. A Llama-8B agent dropped into a population of large cooperating models will defect immediately and pull every partner with it. Two large models playing each other with no intervention will cooperate forever, even in a game whose prompt states the defection equilibrium to both of them. In the clean Direct condition the outcome is symmetric in role order; apparent role-order effects appear only in the higher-variance reasoning conditions and are within their run-to-run spread. None of this is visible if you only evaluate models against themselves.

Two limitations are worth naming directly. The detailed mechanistic analysis (the logit lens and head ablation, Sections 7.1–7.2, and the activation steering and concept clamping, Section 8) uses Llama-3-8B exclusively; the cross-scale and cross-architecture extensions in Sections 7.3 and 7.4 apply the logit lens but do not extend steering to the larger models. The causal claims in this paper therefore rest on Llama-3-8B; the 70B and Qwen results provide suggestive cross-scale and cross-architecture corroboration, but not causal confirmation. We report the multi-seed spread of the self-play cells for Llama-3-8B in Appendix C, and the full self-play grid (mean $\pm$ SD for all four models) in Table 2: the load-bearing results are highly reproducible (the cooperative lock in the three larger models is at $d = 2.00$ with zero variance across seeds), while several chain-of-thought and scratchpad cells are seed-variable and the corresponding claims are scoped accordingly. A remaining limitation is that the round-by-round convergence figure (Figure 1) shows a single representative run per cell rather than a multi-seed average, though the summary statistics it illustrates are reported with multi-seed spread in Table 2. The games are also deliberately simple: two players, two actions. Whether the same circuit structures underlie behavior in richer strategic environments is unknown. The stronger causal evidence comes from activation steering and concept clamping, which demonstrate bidirectional control through a linear direction and confirm it causally encodes the cooperative bias rather than being a spurious correlate. Appendix B reports a control experiment comparing the instruct model to the base pretrained model without RLHF, providing direct evidence on the origin of the cooperative prior.

## 10    Conclusion

We close by summarizing what the results say about LLM strategic behavior, and what questions they open. We set out to understand why LLMs fail to play Nash equilibria. The answer turns out to be more interesting than the question. Where they fail, it is not because they cannot compute the equilibrium: the model we examine internally favors the Nash action and then overrides it. The behavioral failure is a property of the larger models; the 8B we open up actually plays Nash, and it is in that model that we localize and control the override.

Looking inside Llama-3-8B, we found that on contrastive prompts the opponent's action is perfectly separable in the residual stream at every layer, and that the model's Nash preference and the cooperative override that competes with it are visible in the logit lens. Through most of the forward pass, the model privately favors the Nash action. Then, in the final quarter of the network, a distributed cooperative override surges, peaking at about 80% probability of cooperation before the last layer commits back toward the Nash action. Ablating the top opponent-tracking heads leaves the override untouched; it lives in the residual stream more broadly, not in those heads. But it can be steered: a small injection at the mid-to-late layers (20, 22, and 24)

shifts the model's behavior between defection-dominant and cooperation-dominant play, in either direction, without touching any weights.

The behavioral and cross-play experiments establish how this plays out at scale and across architectures. Under direct prompting it is the larger models that fail to play Nash and lock into cooperation, while the 8B plays close to Nash; chain-of-thought moves the larger models toward Nash (with substantial seed variance) and does not worsen the 8B, and we do not claim a clean scale threshold or an architecture ordering for it. The population matters as much as the individual: a single Nash-playing small model makes Nash play contagious in mixed pairings, collapsing the cooperation of any larger partner, while two larger models reinforce each other's cooperation indefinitely. In the clean direct condition these effects are symmetric in role order.

None of this was visible from behavioral experiments alone. It required looking inside. The field has spent years asking why LLMs do not play Nash. We think the more productive question is what generates the late-layer cooperative override, when it prevails behaviorally, and how it can be controlled. The causal answer (activation steering, concept clamping, head ablation) is established for Llama-3-8B. The logit lens results for Llama-3-70B and Qwen2.5-72B show the same phenomenon present across scales and architectures, inviting the extension of causal methods to larger models as a next step.

**Broader Impact Statement**

This work studies whether the strategic behavior of large language models can be understood and controlled at the level of internal representations. The primary application context is the deployment of LLM agents in multi-agent systems such as automated negotiation, auction bidding, or resource allocation. Our findings show that a cooperative prior is present in these models and, in the larger models, is strong enough to suppress Nash equilibrium play behaviorally (the smaller 8B carries the prior internally but plays Nash), and that this prior can be shifted at inference time through activation steering without any retraining.

The potential benefits of this work include enabling more predictable and controllable agent behavior in settings where game-theoretic reasoning matters, and providing interpretability tools that can be applied beyond the strategic domain.

The deployment risks deserve concrete illustration. In multi-round negotiation, an LLM agent with a strong cooperative prior will systematically concede value to a self-interested counterpart, even when the game-theoretically rational strategy is to hold firm. In second-price auctions, cooperative priors may prevent LLM bidding agents from bidding their true valuation, distorting allocation efficiency. In common-pool resource games, a cooperatively biased agent may allow defecting agents to deplete shared resources by failing to adopt protective strategies. In each case, the agent's behavior may appear reasonable in isolation but be exploitable by a strategically sophisticated counterpart.

The steering methodology introduces a symmetric risk: the same technique that can make agents more rational in the above settings could also be used to make them more exploitative in settings where the human counterpart expects cooperation, such as caregiving, counseling, or educational interactions. We make no claim that Nash equilibrium play is the normatively correct behavior in all settings. In many contexts the cooperative prior is desirable and should be preserved. The steering methodology should be applied with care and with explicit awareness of the equilibrium structure and the normative expectations of the deployment context.

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

# A  Prompt Templates

The behavioral experiments use the following prompt structure, applied through the model's chat template. The mechanistic analyses use the same textual structure as a plain string, without the chat template, and with synthetic constructed histories (Section 7.1). Each prompt is constructed from three components: a game description with payoffs, the accumulated round history, and a decision request, all preceded by a fixed system prompt. The reasoning mode (Direct, CoT, Scratchpad) changes only the final instruction line.

**System Prompt**

> You are a rational agent playing a strategic game for real payoffs. Your goal is to maximize your own cumulative payoff over many rounds. Think carefully about your opponent's likely strategy based on past play.

**Game Description**

**Prisoner's Dilemma** (exact frozen prompt; note that the description states the Nash equilibrium, which is the confound discussed in Section 5):

> GAME: Prisoner's Dilemma
> Classic social dilemma. Nash: both defect, but mutual cooperation is Pareto optimal.
> You are Player A. Payoffs shown as (Your payoff, Opponent payoff).
> Actions: Cooperate / Defect
> Payoff matrix:
>   [Opp: Cooperate] | [Opp: Defect]
>   Cooperate: (3,3) | (0,5)
>   Defect: (5,0) | (1,1)

**Battle of the Sexes:**

> You are playing a game called Battle of the Sexes. Each round you choose Opera or Football.
> Payoffs: both choose Opera (2,1), both choose Football (1,2), mismatch (0,0).

**Stag Hunt:**

> You are playing a game called Stag Hunt. Each round you choose Stag or Hare. Payoffs: both hunt Stag (4,4), both hunt Hare (3,3), one hunts Stag alone gets 0, Hare alone gets 3.

**Matching Pennies** (exact frozen prompt; like Prisoner's Dilemma, the description states the equilibrium, the confound discussed in Section 5):

> GAME: Matching Pennies
> Zero-sum game. Unique mixed Nash at 50/50. Tests randomization and exploitation detection.
> You are Player A. Payoffs shown as (Your payoff, Opponent payoff).
> Actions: Heads / Tails

Payoff matrix:
    [Opp: Heads] | [Opp: Tails]
    Heads: (1,-1) | (-1,1)
    Tails: (-1,1) | (1,-1)

We show Prisoner's Dilemma and Matching Pennies as the exact frozen prompts because their descriptions state the equilibrium; the Battle of the Sexes and Stag Hunt descriptions above are given in summary form and do not name an equilibrium.

### History Block

After the game description, the prompt lists the game history:

Game history so far:
Round 1: You played [Action], Opponent played [Action].
Round 2: You played [Action], Opponent played [Action].
. . .

### Decision Instruction by Reasoning Mode

**Direct:**

Round $N$: What do you choose? Answer with exactly one word: [Action1] or [Action2].

**Chain-of-Thought:**

Round $N$: Think step by step about what to do, then state your final answer as exactly one word: [Action1] or [Action2].

**Scratchpad:**

Round $N$: You may reason privately before deciding. Your reasoning will not be shown to your opponent. Then state your final answer as exactly one word: [Action1] or [Action2].

## B  The Cooperative Override Is Rooted in Pretraining, Not in RLHF Fine-Tuning

Throughout the main text we attribute the cooperative override to pretraining on human-generated text, with RLHF partially moderating but not creating the bias. This is a claim, not a proof, and it deserves a direct test. We ran the same logit lens analysis on `meta-llama/Meta-Llama-3-8B`, the base pretrained model that shares identical architecture with `Meta-Llama-3-8B-Instruct` but has never seen instruction tuning or RLHF fine-tuning. If the cooperative override were installed by RLHF, it should be absent or weaker in the base model. It is not.

Looking at the logit lens first (Figure 11), the cooperative override is present in the base model and is in fact stronger than in the RLHF-tuned version. In the instruct model, $P$(Cooperate) peaks at approximately 0.80 in the late layers and the final layer commits to Defect ($P$(Cooperate) $\approx 0.32$). In the base model the cooperative tendency builds earlier, already exceeding 0.60 by layer 13, and peaks near 0.99 at layers 27–28. It never recovers. The base model cooperates completely in its final output, with no partial correction toward the Nash action that the instruct model shows.

The conclusion is straightforward: the cooperative prior is not a product of alignment fine-tuning. It is already there, fully formed, in the pretrained model. The most natural explanation is that the model absorbed it from pretraining on human-generated text, which is predominantly cooperative in tone. What RLHF does is not create the bias but partially restrain it, nudging the final layer back slightly toward Nash without touching the underlying circuit. The cooperative bias, in other words, is not a side-effect of making models helpful and harmless. It is a consequence of learning from human communication itself.

Figure 11: Logit lens comparison: Llama-3-8B-Instruct (RLHF, left) vs Meta-Llama-3-8B (base, no RLHF, right). The cooperative override peaks at $P(\text{Cooperate}) = 0.99$ in the base model versus about 0.80 in the instruct model; in the base model this never corrects in the final layer, whereas the instruct model corrects back to $P(\text{Cooperate}) \approx 0.32$. The override originates in pretraining, not RLHF.

## C  Reproducibility Across Seeds

To assess reproducibility we re-ran every self-play cell across multiple independent random seeds (15 seeds for Llama-3-8B, 10 for the larger models; 50 rounds per run) and report the distribution of the final Nash distance $d^{(50)}$. A confidence interval computed from one 50-round trajectory measures the internal consistency of that trajectory, not whether the summary statistic reproduces; because the rounds of a repeated game are dependent and the per-round behavior drifts over the course of a game (Figure 1), the appropriate unit of variation is the independent run, not the round. The seed-to-seed spread below is that measurement.

Two patterns are worth stating. First, the load-bearing results are highly reproducible. The cooperative lock in the three larger models (Prisoner's Dilemma, Direct mode) is at $d = 2.00$ with zero variance across all seeds, and the near-Nash Direct-mode cells (Prisoner's Dilemma and Stag Hunt for the 8B, and Stag Hunt across models, but not Matching Pennies, where the 8B sits at 0.457) have standard deviations at or near zero. Second, several chain-of-thought and scratchpad cells are genuinely seed-variable: the largest spreads occur for the larger models in Prisoner's Dilemma under reasoning (Qwen-32B CoT $0.44 \pm 0.75$, Qwen-32B Scratchpad $0.84 \pm 0.94$, Qwen-72B Scratchpad $1.34 \pm 0.54$, Llama-70B Scratchpad $0.93 \pm 0.55$). Claims that depend on these cells are scoped accordingly in the main text, and single-run point estimates for them should not be over-interpreted. Table 5 gives the full per-cell spread for Llama-3-8B, the model used for the mechanistic analysis; the complete multi-seed grid for all four models is available with the released results.

## D  Robustness to Decoding Temperature (Greedy, $\tau = 0$)

All results in the main paper use sampled decoding at $\tau = 0.7$, with multi-seed spread reported throughout. As a robustness check on the decoding strategy, we re-ran the full Direct-mode self-play grid (4 models $\times$ 4 games, 50 rounds) with greedy decoding ($\tau = 0$, `do_sample=False`). Greedy generation is deterministic: we verified that repeated runs produce identical action sequences, so a single run per cell is exact rather than an estimate. There were zero parse failures in all 32 agent trajectories. Table 6 reports the final Nash distance $d^{(50)}$ per cell.

Three cells deserve comment, because deterministic decoding produces dynamics that the marginal-based metric summarizes misleadingly, exactly the caveat of Section 5.

First, in Matching Pennies the three larger models each fall into a deterministic synchronized alternation: both agents play Heads, then both play Tails, and so on for all 50 rounds. Because Player A wins on a match, A wins essentially every round (cumulative payoffs $+50/-50$ for Llama-70B and Qwen-72B, $+42/-42$ for Qwen-32B). The per-agent marginals are exactly 50/50, so $d = 0.000$, yet the joint play is maximally

Table 5: Reproducibility of the Llama-3-8B self-play Nash distance across 15 independent random seeds (50 rounds per run). Each row reports the mean and standard deviation of $d^{(50)}$. Direct-mode and near-Nash cells are highly reproducible; several chain-of-thought and scratchpad cells are seed-variable, and the corresponding claims in the main text are scoped accordingly. This multi-seed spread replaces the single-run confidence intervals of the previous version.

| Game | Mode | Mean $d^{(50)}$ | SD | Seeds |
|---|---|---|---|---|
| Prisoner's Dilemma | Direct | 0.045 | 0.010 | 15 |
| | CoT | 0.040 | 0.053 | 15 |
| | Scratchpad | 0.432 | 0.103 | 15 |
| Battle of the Sexes | Direct | 0.014 | 0.019 | 15 |
| | CoT | 0.284 | 0.180 | 15 |
| | Scratchpad | 0.276 | 0.107 | 15 |
| Stag Hunt | Direct | 0.000 | 0.000 | 15 |
| | CoT | 0.292 | 0.222 | 15 |
| | Scratchpad | 0.541 | 0.120 | 15 |
| Matching Pennies | Direct | 0.457 | 0.147 | 15 |
| | CoT | 0.300 | 0.094 | 15 |
| | Scratchpad | 0.206 | 0.120 | 15 |

Table 6: Greedy-decoding ($\tau = 0$) self-play, Direct mode, single deterministic run per cell: final Nash distance $d^{(50)}$. The headline behavioral split of Table 2 reproduces cell-for-cell: Llama-3-8B plays Nash in Prisoner's Dilemma ($d = 0.040$, versus $0.045 \pm 0.010$ at $\tau = 0.7$) while all three larger models lock at full cooperation ($d = 2.000$ exactly). See the text for the Matching Pennies and Battle of the Sexes dynamics, which the marginal-based $d$ does not capture.

| Game | L-8B | L-70B | Q-32B | Q-72B |
|---|---|---|---|---|
| Prisoner's Dilemma | 0.040 | 2.000 | 2.000 | 2.000 |
| Battle of the Sexes | 0.000 | 0.000 | 0.333 | 0.000 |
| Stag Hunt | 0.000 | 0.000 | 0.000 | 0.000 |
| Matching Pennies | 0.780 | 0.000 | 0.000 | 0.000 |

exploitative: this is the sharpest illustration in our data that $d$ measures distributional proximity to an equilibrium, not strategic coherence. Greedy decoding cannot implement a mixed strategy at all, so the Matching Pennies equilibrium (randomize 50/50) is unreachable by construction at $\tau = 0$; the synchronized alternation is the deterministic surrogate the models adopt. This is the structural reason the main experiments use sampled decoding.

Second, Llama-3-8B in Matching Pennies begins in the same synchronized alternation, desynchronizes around round 6, and settles into a fixed exploited pattern (its opponent wins the remaining rounds; cumulative payoff $-32/+32$), giving $d = 0.780$.

Third, Qwen-32B in Battle of the Sexes alternates between the two equilibria but perfectly out of phase with its identical partner: the two agents mismatch on all 50 rounds and both receive a total payoff of exactly 0.0, while each agent's marginal is 50/50 ($d = 0.333$). Deterministic decoding here locks in permanent anti-coordination, whereas at $\tau = 0.7$ this cell coordinates.

Everywhere else the picture matches the main grid: all four models coordinate on Stag in Stag Hunt ($d = 0.000$), and three of four lock onto Opera in Battle of the Sexes. The central Prisoner's Dilemma finding, the scale-dependent split between the Nash-playing 8B and the cooperatively locked larger models, is unchanged under deterministic decoding.

## E   Model Outputs Under Intervention

The steering and clamping quantities in Section 8 are probabilities renormalized over the two action tokens at the decision position: the model is run on the prompt (optionally with the intervention applied), the final-position logits are read, and the probability mass on Cooperate versus Defect is compared. They are properties of the decision readout, not of free-running generation. This appendix documents generation directly: greedy continuations (8 tokens) from the contrastive prompts, under the same saved steering vector used for the published sweep, with steering applied at layers 20, 22, and 24 at all positions (as in the main experiments) and clamping applied at the final position of layer 22 (as in the main experiments).

As a faithfulness check, the published readout curve reproduces on freshly constructed prompts under the saved vector: baseline renormalized $P(\text{Defect}) = 0.618$ (recorded sweep: 0.616); $\alpha = -5$ gives $P(\text{Defect}) = 0.995$ (recorded: 0.992); clamping at $c = -30$ and $c = +30$ gives $P(\text{Cooperate}) = 0.000$ and 0.914 (recorded: 0.001 and 0.986). The direction therefore transfers across prompt samples.

Table 7: Greedy 8-token continuations under intervention (two prompts per condition produced identical or near-identical text; one shown). Text is verbatim except that newlines are shown as spaces and "[non-ASCII ×8]" abbreviates a repeated non-Latin token. The readout column is the renormalized $P(\text{Cooperate})$ measured in the same run.

| Intervention | Value | $P(\text{Coop})$ | Greedy continuation |
|---|---|---|---|
| Steering | $\alpha = -20$ | 0.042 | [non-ASCII ×8] |
| Steering | $\alpha = -5$ | 0.005 | `. My response:   (to be filled` |
| None | $\alpha = 0$ | 0.382 | `Please respond with your answer.   I` |
| Steering | $\alpha = +10$ | 0.816 | `BUT COBO COBO` (degraded) |
| Steering | $\alpha = +40$ | 0.953 | [non-ASCII ×8] |
| Clamping | $c = -30$ | 0.000 | `. (no) (no) (` |
| None | $c = 0$ | 0.394 | `Please respond with your answer.   I` |
| Clamping | $c = +30$ | 0.914 | `Please make a decision, and we` |

The pattern is systematic and tracks the structure of the two interventions. Clamping, which modifies only the final position, preserves fluent and well-formed text across the entire swept range, including at the cooperative extreme ($c = +30$, where the readout flips decisively to cooperation and the continuation remains ordinary instructional English). Steering, which injects $\alpha \mathbf{v}_{\text{coop}}$ at every sequence position, preserves fluent text at the moderate defect-side operating point ($\alpha = -5$) but degrades free-form fluency at $\alpha = +10$ and collapses it at the swept extremes, even as the decision-position readout remains sharp and directionally consistent throughout. This is the expected signature of an all-position injection: a large added vector perturbs the representation of the entire context, while the relative ordering of the two action tokens at the decision position, which is precisely the axis the direction encodes, remains reliable. Two scope conclusions follow. The quantitative steering and clamping claims of Section 8 are claims about the decision readout, as defined there. And causal control of the cooperative override with fully fluent generation is demonstrated by the clamping intervention; the extreme ends of the steering sweep should be read as stress tests of the direction, not as operating points for deployed generation.

## F   Prompt-Surface Ablation: What Carries the Effects?

Reviewer discussion raised the concern that the four canonical games are present in training data and that the prompts name the games (and, for two games, state the equilibrium), so that the behavioral results might reflect recognition of familiar surface forms rather than responses to payoff structure. This appendix tests that concern directly on the Prisoner's Dilemma structure with a five-condition prompt-ablation ladder that varies the surface while holding the game fixed up to an affine payoff rescaling (which preserves the equilibrium structure exactly). Table 8 gives the design; every condition is run in self-play, Direct mode, 50 rounds, with one greedy run ($\tau = 0$) and three sampled runs ($\tau = 0.7$; five for the fully disguised condition) per model.

Table 8: The five conditions. "Hint" is the sentence in the game description stating that mutual defection is the Nash equilibrium. Rescaled payoffs are the affine transform $2x + 1$ of the originals (7, 1, 11, 3 in place of 3, 0, 5, 1), which preserves best responses and equilibria exactly.

| Condition | Framing | Hint | Payoffs | Action labels |
|---|---|---|---|---|
| Named (main paper) | "Prisoner's Dilemma" | yes | original | Cooperate/Defect |
| Hint-ablated | "Prisoner's Dilemma" | no | original | Cooperate/Defect |
| Unnamed | neutral task | no | original | Cooperate/Defect |
| Unnamed, rescaled | neutral task | no | rescaled | Cooperate/Defect |
| Fully disguised | neutral task | no | rescaled | Circle/Square |

Table 9: Cooperation rate of Player A (probability mass on Cooperate, or on Circle in the fully disguised condition) at $\tau = 0.7$, mean $\pm$ SD over independent runs, with the single greedy ($\tau = 0$) run in parentheses. Named-condition values are the main paper's (Table 2, 15/10 seeds).

| Condition | L-8B | L-70B | Q-32B | Q-72B |
|---|---|---|---|---|
| Named | 0.02 | 1.00 | 1.00 | 1.00 |
| Hint-ablated | $0.03 \pm 0.01$ (0.02) | $1.00 \pm 0.00$ (1.00) | $1.00 \pm 0.00$ (1.00) | $0.91 \pm 0.13$ (1.00) |
| Unnamed | $0.51 \pm 0.39$ (1.00) | $1.00 \pm 0.00$ (1.00) | $1.00 \pm 0.00$ (1.00) | $1.00 \pm 0.00$ (1.00) |
| Unnamed, rescaled | $1.00 \pm 0.00$ (1.00) | $1.00 \pm 0.00$ (1.00) | $1.00 \pm 0.00$ (1.00) | $0.89 \pm 0.16$ (1.00) |
| Fully disguised | $0.72 \pm 0.32$ (1.00) | $0.52 \pm 0.13$ (0.74) | $0.36 \pm 0.29$ (0.08) | $0.47 \pm 0.21$ (1.00) |

Three findings follow, one per ablated factor.

First, the explicit equilibrium statement is behaviorally inert. Removing only the hint (row 2 versus row 1) changes no cell: the 8B still defects almost every round and the larger models still lock at full cooperation. The disclosed prompt leakage of Section 4 does not drive the results: the 8B defects without being told the equilibrium, and the larger models cooperate despite being told.

Second, the action labels carry both the behavioral lock and the mechanistic signature. Across the first four rows, wherever the labels are Cooperate and Defect, the three larger models cooperate at or near 100% regardless of framing, hint, or payoff values; replacing only the labels with Circle and Square (row 5 versus row 4, a single-factor change) dissolves the lock into noisy mixed play in all four models. The logit lens shows the same gate internally (Figure 12): on the four Cooperate/Defect conditions the 8B's layer profile is essentially identical, with the late cooperative surge peaking at layer 29–30 (0.93–0.95 on this fresh prompt sample) and the final-layer correction toward Defect (final 0.36–0.39), whereas under Circle/Square the late surge is absent altogether: the Circle-analog peaks at layer 8 (0.99) and declines monotonically to 0.15 at the final layer. The cooperative override of Sections 7.1 and 8 is engaged by the cooperation vocabulary itself. This is consistent with the mechanism the paper describes, a prosocial direction whose readout is the Cooperate token, and it identifies the override's trigger as lexical rather than structural.

Third, the 8B's near-Nash play is keyed to the game framing, not the payoff matrix. On the identical matrix (3, 0, 5, 1) presented as a neutral decision task (row 3), the 8B flips from 2% to 100% cooperation under greedy decoding, and to unstable mixed play under sampling ($0.51 \pm 0.39$); its internal lens profile, measured on synthetic Nash-history prompts, is meanwhile unchanged from the named condition (peak 0.93 at layer 30, final 0.37). Its defection policy therefore tracks recognition of the named game rather than a computation over the payoffs, and the dissociation between the unchanged decision-position readout and the flipped sampled behavior in this condition is a further instance of the readout-versus-generation distinction of Appendix E.

The scope implications are stated in Section 9: the paper's claims concern the canonical games as presented, the leakage confound is empirically retired for the equilibrium hint, and equilibrium computation from payoff structure alone is explicitly not established for any model tested. Surface dependence of this kind is itself a finding the framework makes measurable, and mapping it beyond the Prisoner's Dilemma structure is natural future work.

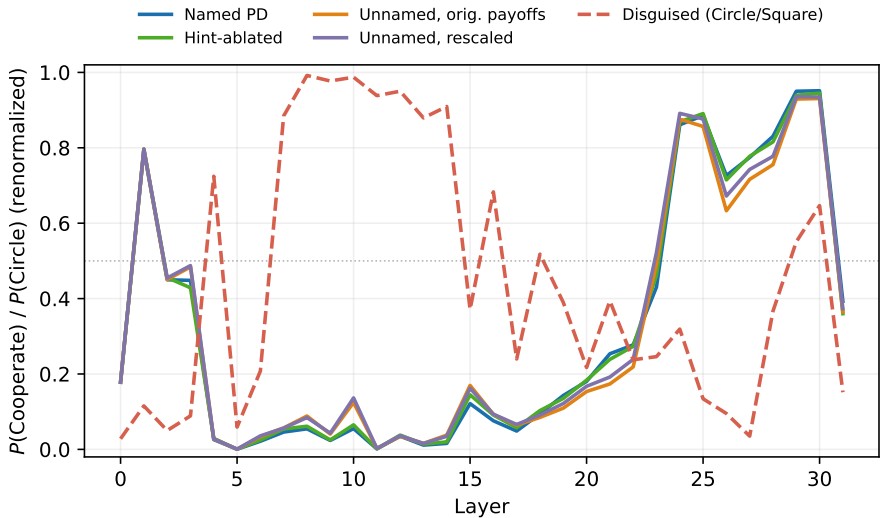

Figure 12: Logit lens on Llama-3-8B across the five prompt conditions (renormalized two-token readout at the decision position; 20 synthetic Nash-history chain-of-thought prompts per condition, fresh sample). The four Cooperate/Defect conditions are essentially indistinguishable, late surge and final-layer correction included; the fully disguised condition (Circle/Square, dashed) shows no late surge at all. Peak magnitudes on this fresh sample run higher than Figure 2's (0.95 versus 0.80), the prompt-sample sensitivity documented in Appendix E; the within-run comparison across conditions is the object here.

