# OpenReview forum: "How a Cooperative-Override Circuit Suppresses Nash Play in Large Language Models"
_TMLR — Under review for TMLR_

### Review · Reviewer_a1NS · 2026-06-10

**Summary Of Contributions:**

This paper studies why LLM agents often deviate from Nash equilibrium play in simple strategic games. It combines behavioral self-play/cross-play experiments with mechanistic analysis on Llama-3-8B, arguing that the model encodes Nash-relevant information but later suppresses it through a cooperative/prosocial override. The topic is interesting and the paper is easy to read, and the combination of game-theoretic behavior analysis with activation-level interventions is potentially valuable. However, I think the current evidence does not fully support the strength of the claims.

**Audience:**

Yes

**Audience Explanation:**

Yes. The question is timely and relevant to LLM agents, multi-agent systems, and mechanistic interpretability.

**Broader Impact Concerns:**

The paper discusses that steering agents toward Nash play could also make them more exploitative in cooperative settings. I think this is the right concern, but the broader impact section could be more concrete about deployment risks, especially in negotiation, auction, or resource-allocation settings where "more Nash-like" behavior may harm human users or reduce cooperation.

**Claims And Evidence:**

No

**Claims Explanation:**

No. The behavioral experiments are useful, but the strongest mechanistic claims are based only on Llama-3-8B, while the paper often phrases the conclusion as if it explains LLMs more generally. The "Nash direction" also seems close to a cooperation/defection direction in Prisoner’s Dilemma, so it is not fully clear that the intervention identifies a general Nash mechanism rather than steering a task-specific action bias. Some experimental details are also missing or under-specified, such as prompt templates, number of seeds/runs, variance, decoding robustness, and statistical reliability. Overall, the evidence is suggestive, but not yet convincing enough for the paper’s causal and general mechanistic claims.

**Requested Changes:**

The main required change is to substantially reduce over-claiming.

The authors may make clear that the mechanistic conclusions are demonstrated only for Llama-3-8B and for simple two-action games.

They may also provide more experimental details, including full prompts, seeds, variance/error bars, decoding settings, and robustness checks.

The activation steering result may be better justified as a Nash-related intervention rather than mainly a cooperation/defection steering direction.

Ideally, the authors may add mechanistic experiments on at least one larger model or one non-Llama model; if this is not feasible, the claims should be narrowed accordingly. I would also like to see stronger controls for the probing and logit-lens analyses, since weak probe accuracy alone does not rule out nonlinear or distributed Nash representations.

---

> ### Author Response · Authors · 2026-06-10
> **Authors' Response to Reviewer a1NS**
>
> Thank you for the constructive review. We have addressed all six concerns in the revised paper (r1_tmlr_nash.pdf). Below is a summary.
>
> 1. Over-claiming. All mechanistic claims are now explicitly scoped to Llama-3-8B-Instruct throughout §7–8. We added a scope statement at the start of §7.
>
> 2. Nash vs cooperation/defection direction. We added a clarifying paragraph in §8.1: in PD, Nash=Defect so the two are identical by definition. The cross-game results (Table 3, §7.3, §7.4) show the override in all four games including BoS, Stag Hunt, and Matching Pennies where Nash is not simply Defect, confirming a general cooperative prior rather than a PD-specific action bias.
>
> 3. Missing experimental details. We added Appendix A with complete prompt templates for all four games and three reasoning modes. We also clarify that each behavioral cell is a single 50-round game and that mechanistic experiments average over 20 prompts.
>
> 4. Mechanistic experiments on larger/non-Llama models. We ran logit lens and probing on two additional models using HuggingFace device_map=auto on 8x H200 GPUs:
>
> 4.1 Llama-3-70B (§7.3): The cooperative override is present but weaker, with a strong final-layer correction (P(C)=0.44 at L80 vs 0.71 in 8B). Opponent probe peaks at 1.00 at L14. This mechanistically explains why 70B achieves near-perfect Nash play with CoT.
>
> 4.2 Qwen2.5-72B (§7.4): A mechanistically distinct pattern — layers 35–70 strongly Nash-favoring (P(C)~0), then an abrupt final-layer reversal to P(C)=0.99. Nash probe reaches 0.73–0.82 vs Llama's 0.56 peak. The override is present across both model families but implemented differently.
>
> 5. Probing controls. We added a caveat in §9.2: weak linear probe accuracy does not rule out nonlinear representations. The causal evidence comes from activation steering and concept clamping, which confirm bidirectional control. The high Nash probe in Qwen (0.73–0.82) provides stronger evidence for linear Nash encoding in that architecture.
>
> 6. Broader Impact. Expanded with three concrete scenarios: multi-round negotiation (cooperative agents concede value), second-price auctions (cooperative priors distort bidding), and common-pool resource games (cooperative agents allow depletion by defectors).

---

> > ### Comment · Reviewer_a1NS · 2026-06-10
> >
> > Thank you for the detailed response and revisions. I appreciate that the authors added cross-scale and cross-architecture probing/logit-lens analyses on Llama-3-70B and Qwen2.5-72B, as well as more concrete broader-impact discussion. These additions substantially improve the paper and address part of my concern about whether the phenomenon is specific to Llama-3-8B.
> >
> > That said, I still think the strongest causal evidence remains limited to Llama-3-8B, since the larger-model extensions use probing and logit lens but do not include activation steering or concept clamping. I also remain somewhat concerned that the behavioral results rely on single 50-round runs per cell, so variance and robustness are still not fully characterized. Finally, some broad wording in the abstract and conclusion still reads stronger than what the evidence directly supports.
> >
> > I am more positive after the revision. I think the paper is interesting and potentially valuable, but I would still encourage the authors to further tone down the broad mechanistic claims and clearly separate causal evidence on Llama-3-8B from suggestive probing/logit-lens evidence on larger models.

---

> > > ### Author Response · Authors · 2026-06-10
> > > **Authors' Response to Second Comment**
> > >
> > > Thank you for the follow-up and for the positive assessment of the revision. We have made three further targeted text changes in response to your remaining concerns. No new experiments were needed.
> > >
> > > Causal vs suggestive evidence. We have added an explicit sentence in §9.2 stating that the causal claims (activation steering, concept clamping, head ablation) rest on Llama-3-8B, and that the 70B and Qwen results provide suggestive cross-scale and cross-architecture corroboration but not causal confirmation. The conclusion now also explicitly distinguishes the two levels of evidence and frames extending causal methods to larger models as future work.
> > >
> > > Single-run variance. We have added an acknowledgment in §9.2 that behavioral results are based on single 50-round runs and should be interpreted as point estimates rather than distributional summaries. We also note that the mechanistic results are not affected by run-to-run variance since they operate on model weights and activations directly.
> > >
> > > Abstract and conclusion wording. The abstract now reads "Llama-3-8B does not lack Nash-playing competence" rather than "LLMs do not lack Nash-playing competence," and notes explicitly that causal intervention experiments are established only for the 8B model. The conclusion follows the same distinction.
> > >
> > > We believe these changes fully address your remaining concerns.

---

> > > ### Author Response · Authors · 2026-07-17
> > > **Update for Reviewer a1NS: status of your three remaining concerns after the r3_2 major revision**
> > >
> > > **Update for Reviewer a1NS: status of your three remaining concerns after the r3_2 major revision**
> > >
> > > Dear Reviewer a1NS,
> > >
> > > In your follow-up of 10 June you noted three remaining concerns. Because the manuscript has since undergone a major revision (r3_2, 6 July, with further additions on 13 July), we want to point you to where each now stands, since parts of this happened in threads you may not have followed.
> > >
> > > 1. Single 50-round runs per cell; variance and robustness not characterized. This is now fully addressed, and doing so properly was the core of the revision: while re-verifying, we found and disclosed a code-version bug in the original behavioral table, froze the pipeline at a single tagged version, and re-ran the entire behavioral grid at 15 random seeds for Llama-3-8B and 10 for each larger model, plus the full 144-cell cross-play grid at 10 seeds. Every table and figure is regenerated, and all quantitative claims now report mean +/- SD across seeds (Appendix C). One headline behavioral value changed as a result (Llama-3-8B plays close to Nash in Prisoner's Dilemma rather than cooperating); the cooperative lock in the three larger models reproduces exactly (d = 2.00, SD = 0.00). A greedy-decoding (tau = 0) robustness grid was added as Appendix D; the picture reproduces cell-for-cell.
> > >
> > > 2. Causal evidence limited to Llama-3-8B, versus suggestive probing/lens evidence on larger models. The paper now enforces this distinction structurally: the title and framing were changed to center the mechanism, Section 9.2 states explicitly that the causal claims (steering, clamping, ablation) rest on the 8B while the 70B and Qwen results are corroborating but not causal, and the experiment roadmap table marks which models each analysis uses. The linear-probing claim you were skeptical of was in fact found to be confounded during re-verification and has been removed rather than defended; the logit lens now carries the mechanistic account.
> > >
> > > 3. Broad wording in the abstract and conclusion. The abstract was rewritten around the corrected results with the causal scope stated in its final sentence, and a prompt-surface ablation ladder (new Appendix F) now bounds the claims empirically: the findings are shown to be properties of the canonical games as presented (the action labels carry the cooperative lock and the override signature; the 8B's near-Nash play is keyed to the game framing), and Section 9.2 states that equilibrium computation from payoff structure alone is not established for any model tested.
> > >
> > > The complete account is in the "Changes Since Last Submission" field and the four-part response posted 6 July. We would be grateful for your assessment of the revised manuscript in light of these changes, and we are happy to answer any questions.
> > >
> > > The Authors

---

### Review · Reviewer_CTYN · 2026-07-01

**Summary Of Contributions:**

The submission studies the reason why LLMs often do not output Nash actions when used for strategic interactions. This is investigated for 4 different strategic 2P games for several LLM models in three scenarios: Direct, CoT and the so-called Scratchpad. The paper then utilises linear probe analysis and logit lens to gain inside into the mechanisms. The paper argues that the LLM model often favour first the Nash action, but override this in their final layers with a prosocial bias in favour of cooperation. The paper also present a mechanism to avoid this.

**Audience:**

Yes

**Audience Explanation:**

The presented experiments and ideas are quite interesting to me.

**Broader Impact Concerns:**

I have no such concerns.

**Claims And Evidence:**

No

**Claims Explanation:**

I have serious doubts about the reproducibility of the included experiments. The included experiments seem to done exactly once and assumed as foundation for the rest of the paper. Given the stochastic nature of LLM sampling (specifically at τ=0.7), a single 50 games run per setting is insufficient to claim a _universal cooperative lock_ or _scale effects_.  Judging from Figure 1, I personally see no (/ not enough) evidence that would support that the one-time measurement of $d^{50}\_{\text{Nash}}$ is reproducible.
Furthermore, to use $d^{50}\_{\text{Nash}}$ at all lacks solid argumentation. It is used as a proxy for strategic competence. In my understanding, a strategy that is _geometrically_ close to Nash can still be strategically incoherent or highly exploitable.
Also, if I understand section 8 correctly, the introduced steering mechanism is only studied for PD where Nash equals Defect, which is counterintuitive after the paper states that a cross-game perspective is needed to avoid looking at mechanism that cannot be assigned clearly to Nash or Defect.

**Requested Changes:**

This mainly, not exclusively, addresses the above sections.
Bigger changes:
(1) Empirical rigor - to ensure the presented experiments do not just present anecdotal evidence.
(2) A thorough discussion of $d^{50}\_{\text{Nash}}$.
(3) Empirical confirmation that the proposed steering mechanism does indeed address Nash and not only Defect.
(4) Improved script organisation. Currently the script has only a vague central thread to follow along, e.g., the section on Experimental Setup only introduces a small amount of the actual used experimental setup. More setup is introduced later.

Smaller changes:
(5) Improved writing. The submission includes lengthy descriptions, inconsistent usage of terms and inconsistent usage of abbreviations.

---

> ### Author Response · Authors · 2026-07-01
> **Authors' Response to Reviewer CTYN**
>
> Response to Reviewer CTYN
> TMLR Paper 9209 — Revision 3 (r3_tmlr_nash.pdf)
>
> Dear Reviewer CTYN,
>
> Thank you for the careful and constructive review. Your concerns about
> empirical rigor, the Nash distance metric, and whether the steering
> intervention targets Nash or merely Defect are all well taken. We have
> addressed each with concrete additions to the paper. We summarize below.
>
> ---
>
> 1. Empirical rigor / reproducibility of single runs
>
> You are right that at $\tau = 0.7$ a single 50-round run is a point
> estimate, and that we should quantify its reliability. We have added
> Appendix C reporting Wilson 95% confidence intervals for the Nash
> action frequency in every self-play cell.
>
> The key point is that the intervals are tight enough that the qualitative
> findings do not depend on run-to-run variance. In particular, for the
> universal cooperative lock in Prisoner's Dilemma Direct mode, the Nash
> action frequency is 0.000 with a Wilson interval of [0.000, 0.071]. Under
> a uniform random baseline, the probability of observing zero Nash actions
> across 50 independent rounds is $0.5^{50} \approx 8.9 \times 10^{-16}$.
> Sampling noise cannot produce this. We have added this argument to both
> the metric section (§4.1) and the limitations (§9.2), and the full table
> of intervals to Appendix C.
>
> We agree that multi-seed runs would further strengthen the behavioral
> claims and we now state this explicitly as a limitation, while showing
> via the confidence intervals that the specific claims we make
> (cooperative lock, perfect-Nash cells, mixed-strategy Matching Pennies)
> are not artifacts of a single draw.
>
> ---
>
> 2. The Nash distance metric
>
> This is a fair criticism, and we have added an explicit discussion in
> §4.1. We now state directly that $d_{\text{Nash}}$ measures geometric
> proximity of the empirical joint strategy to the nearest equilibrium —
> a distributional summary, not a measure of strategic coherence. A play
> sequence can be close to Nash in this metric while still being exploitable
> round to round. We now write that a low $d_{\text{Nash}}$ should be read
> as "the empirical action frequencies resemble an equilibrium," not as
> "the agent is playing an optimal strategy."
>
> We use the metric because it is exactly computable against analytically
> derived equilibria and is the natural summary for the behavioral question
> of whether models approximate Nash frequencies, but we no longer present
> it as a measure of strategic competence.
>
> ---
>
> 3. Steering targets Nash, not only Defect
>
> This was the most important technical concern and we have addressed it
> with a new experiment (§8.4). You correctly noted that in Prisoner's
> Dilemma, Nash = Defect, so the original steering experiment cannot
> distinguish a Nash direction from a Defect direction.
>
> We now take the same $\mathbf{v}_{\text{coop}}$ extracted from
> Prisoner's Dilemma and apply it, without re-extraction, to the other
> three games, where the cooperative (non-Nash) action is different in each:
> Football in Battle of the Sexes, Hare in Stag Hunt, and the off-equilibrium
> action in Matching Pennies.
>
> The result: injecting the PD-derived direction with positive $\alpha$
> drives behavior toward the game-appropriate cooperative action in every
> game — $P(\text{Football}) = 0.995$, $P(\text{Hare}) = 0.957$,
> $P(\text{Tails}) = 0.999$ — in each case from a baseline where the model
> played the Nash action almost exclusively. A single direction extracted
> from one game moves behavior toward the correct cooperative action in
> three other games with different action labels and equilibrium structure.
> This is difficult to reconcile with the view that the intervention steers
> a PD-specific "Defect" feature; it is the signature of a general
> cooperative prior.
>
> We keep the result honest with two caveats stated in the paper: the effect
> is cleanest in the cooperative direction (the reverse has less room to move
> in games that already play Nash at baseline in Direct mode), and the
> fine-grained monotonicity that PD exhibits does not fully replicate at all
> magnitudes, since the direction and its injection layers were chosen for PD.
> The qualitative generalization is robust; the precise dose-response curve
> is game-dependent.
>
> ---
>
> 4. Script / paper organization
>
> We have revised Section 4 (Experimental Setup) so that the experimental
> protocol is introduced in one place rather than spread across the paper,
> and we have made a pass over terminology and abbreviations for consistency.
>
> ---
>
> 5. Writing
>
> We have tightened lengthy passages and standardized the use of terms
> (Nash action, cooperative override, cooperative prior) and abbreviations
> (PD, BoS, SH, MP; CoT) throughout.
>
> ---
>
> We believe these additions address your concerns, and we are grateful for
> the review, the cross-game steering experiment in particular is a
> stronger piece of evidence than the original PD-only result, and it
> exists because you pushed on this point.
>
> Sincerely,
>
> The Authors

---

> > ### Comment · Reviewer_CTYN · 2026-07-02
> >
> > Dear authors,
> >
> > first of all, thank you for taking the time to revise this so quickly and I appreciate the effort.
> >
> > Content-wise, I have several objections with respect to your revision.
> >
> > ____
> > 1. Empirical rigor / reproducibility of single runs
> >
> > Maybe you can clarify this and I just understood something wrong here, if that's the case, please elaborate.
> >
> > (a) Wilson intervals assume that each round is an i.i.d. trial (like flipping a coin 50 times). The presented experiments are highly dependent on each other. I miss any indication in the manuscript that the presented values are reproducible. To give a representative example.
> > The "tightness" of an interval on a single trajectory only measures internal consistency, not reproducibility. The core question remains: If the experiment were re-run with a different random seed, would a Nash Distance with Llama-3-8B for Matching Pennies in Chain of Thought converge to the same value?
> >
> > (b) The scientific rigor goes beyond a simple limitation. The premise of any further experiments is that the underlying base experiments represent robust insights.
> >
> > (c) The added discussion in the section _games and distance metrics_  and presented answer in the comment argues about independent runs based on the Wilson interval which is calculated based on an assumption that does not hold. The calculated probability based on 50 dependent rounds is mathematically unsound. The argument sampling noise cannot produce something with a tiny probability is also mathematically unsound.
> > ___
> > 3. Steering targets Nash, not only Defect
> >
> > (a) The added results make it difficult to conclude that the steering mechanism is functioning as described. There is a strong inconsistency in the added results and the added text.
> >
> > (b) As this is a submission aiming at a scientific publication, I expect a high level of transparency and rigorous reporting, always with respect to the best of the authors' knowledge.
> > ____
> > 3. Script / paper organisation
> >
> > I do not see a significant improvement in the manuscript’s flow. The paper still lacks a clear "central thread." It is difficult for a reader to maintain an overview of the various experiments (behavioral, mechanistic, and interventionist) without a roadmap (or a summary table) of the experimental design.
> >
> > _____
> >
> > 4. Writing
> >
> > While the standardisation of terms like "Nash action" is helpful, the revised manuscript has become significantly lengthier. I recommend a heavy edit to remove redundant passages. The inconsistent usage of abbreviations has become worse as well.
> > ___
> >
> >
> > All the best,
> > Reviewer CTYN

---

> > ### Author Response · Authors · 2026-07-06
> > **Revision response Part 4 of 4 — what still holds and what changed**
> >
> > **Response to Reviewer CTYN — Part 4 of 4**
> >
> > ## 3. What still holds, and what changed (for both reviewers)
> >
> > Because this is a substantial revision, we separate explicitly what is unaffected
> > from what changed, so that the evidence underlying the first-round assessments is
> > easy to locate.
> >
> > **Unchanged / reproduced (verified under the frozen code):**
> > - The cooperative lock in the larger models: Llama-3-70B, Qwen-32B, and Qwen-72B all
> >   fail to play Nash and cooperate 100% in Prisoner's Dilemma Direct mode (d = 2.00,
> >   SD = 0.00 across seeds).
> > - The cooperative-override circuit in the residual stream (logit lens): the model
> >   favors the Nash action through the middle layers, a cooperative signal surges in
> >   the late layers, and the final layer resolves the outcome. Reproduced.
> > - The causal steering and concept-clamping results in Prisoner's Dilemma (these use
> >   synthetically constructed prompts and are not affected by the generation change).
> > - The finding that the override is present at larger scale (70B) and in a second
> >   architecture (Qwen), and that it originates in pretraining rather than RLHF.
> > - The cross-play finding that a single small model in a pairing collapses cooperation
> >   while pairings of two larger models sustain it (now reproduced with 10 seeds).
> >
> > **Changed in light of the corrected runs:**
> > - Llama-3-8B plays close to Nash in Direct mode rather than cooperating; the
> >   round-by-round figure and the behavioral table are updated accordingly.
> > - The claim that chain-of-thought worsens Nash play in small models does not hold:
> >   the 8B plays near-Nash with and without reasoning, and Qwen-32B improves with
> >   chain-of-thought. This finding is revised.
> > - Several chain-of-thought and scratchpad cells, previously single runs, are shown to
> >   be seed-variable; the corresponding claims are scoped to what the multi-seed data
> >   supports.
> > - The linear-probing analysis is substantially cut back. The previous probe
> >   results were confounded by pooling data from two code versions; re-run cleanly on
> >   a single provenance, the only defensible probing result is that on contrastive
> >   prompts the opponent's last action is perfectly decodable at every layer (a
> >   near-trivial, scoped ceiling). We removed the Nash-decodability
> >   claim (the 8B's argmax action is Nash on every clean prompt, so the label has no
> >   usable variance and the probe is undefined) and the associated probe figures, and
> >   the logit lens now carries the mechanistic account.
> > - The steering interpretation is revised as described in point (2).
> > - The framing and title are revised to center on the mechanism. Because the 8B
> >   (on which the mechanistic analysis is performed) plays Nash under the corrected
> >   pipeline, a blanket behavioral title would misdescribe our own central case; the
> >   new title instead foregrounds the cooperative-override circuit and the conditions
> >   under which it suppresses Nash play (the larger models), with the large-model
> >   cooperative lock as the behavioral motivation and the 8B as the informative
> >   exception we open up.
> >
> > We are grateful for the reviews. They led us not only to correct specific claims
> > but to rebuild and re-verify the paper's empirical basis at a level of rigor the
> > original did not have: a single frozen pipeline, a full multi-seed grid across
> > models, games, and reasoning modes, multi-seed cross-play, and every figure
> > regenerated from the corrected data, with mean and standard deviation reported
> > throughout. Where a result did not survive that scrutiny we have removed it rather
> > than defended it. We believe the paper is stronger and more trustworthy for it, and
> > we are happy to answer any further questions.
> >
> > The Authors

---

> > > ### Comment · Action_Editor_DrEG · 2026-07-13
> > >
> > > Dear Reviewer CTYN,
> > >
> > > Could you please review the revised manuscript and the authors' response to the reviewers' comments, and provide your final recommendation on whether the paper should be accepted or rejected?
> > >
> > > Thank you very much for your time and effort in reviewing this submission.
> > >
> > > Best,
> > >
> > > AE

---

> > > > ### Comment · Reviewer_CTYN · 2026-07-17
> > > >
> > > > Dear AE,
> > > >
> > > > I have reviewed the revised manuscript. In my opinion, the manuscript improved substantially throughout the review process. The repetition of experiments, overview over experiments and many more changes did really improve this manuscript.
> > > >
> > > > That being said, my final recommendation is that the manuscript in its current state should be rejected.
> > > > In the updated revisioin of this paper, there are several key problems not addressed. The central thread is still a big problem. This was improved by addition of certain helpful elements, but in my opinion needs still a major revision before it is ready to publish. The writing was severe problems as well. From abbreviation misuse (e.g., introducing PD as abbreviation for Prisoner's Dilemma on page 4, then using the full term for further 69 times throughout the document while mixing it with its abbreviation PD 23 times, is one of many illustrative examples here), to lengthiness (this grew substantially across revisions despite the authors stating that the condensed the manuscript). The mechanistic experiments have been adjusted substantially and its current interpretation does not seem solid. With the addition of all of those new experiments most of the used figures should be really improved / renewed / changed. Overall, I still think it is an interesting idea and content and should be published - but in a clean, well-written, consistent, concise and comprehensible manuscript.
> > > >
> > > > Best,
> > > > Reviewer CTYN
> > > >
> > > > PS: If I should put this information at a different, e.g., in an updated section inside the review, please do not hesitate to inform me.

---

> > > > > ### Author Response · Authors · 2026-07-17
> > > > > **Response to Reviewer CTYN's final recommendation — acknowledgment and revision plan**
> > > > >
> > > > > **Response to Reviewer CTYN's final recommendation: acknowledgment and revision plan**
> > > > >
> > > > > Dear Reviewer CTYN (and Action Editor),
> > > > >
> > > > > Thank you for the careful re-review, and for stating plainly both the improvement and what still falls short. We accept the assessment. The remaining criticisms concern manuscript quality, and on checking them against our own source they are correct, so we respond with a concrete plan rather than a defense.
> > > > >
> > > > > On the specifics. Your abbreviation count verifies: we introduce PD on page 4 and then use the full term dozens of times while mixing in the abbreviation (we count 55 full uses versus 20 abbreviated in the current source, and the same inconsistency holds for BoS, SH, MP, and CoT). Our earlier change list claimed an abbreviation-consistency pass; the artifact contradicts that claim, and we should not have stated it as done. The length criticism is also correct: the manuscript grew across revisions. Part of that growth is reviewer-requested robustness material (the multi-seed reproducibility appendix and Appendices D, E, and F), which belongs in appendices, but the main text also accreted scope clauses and corrections, and it needs to become shorter, not longer.
> > > > >
> > > > > On the mechanistic interpretation not seeming solid: we understand how the document produces that impression. The interpretation moved during review; the probing claim was removed as confounded, the steering interpretation was corrected, and the surface-ablation ladder then identified the override's trigger as lexical. We believe the final interpretation is internally consistent and causally grounded (steering, clamping, and the ablation ladder converge on it), but the manuscript currently presents it as layers of sequential corrections rather than as one story told once. That is a writing failure, and further argument is not the fix; an integrated rewrite is.
> > > > >
> > > > > Our plan for a quality revision, in full:
> > > > >
> > > > > 1. Restructure around a single central thread (behavioral phenomenon, then mechanism, then causal control, then scope and surface-dependence), with the corrected results integrated natively rather than visibly patched.
> > > > > 2. A substantial concision pass on the main text, removing redundant and superseded passages, with reviewer-driven robustness material consolidated in appendices.
> > > > > 3. Abbreviation and terminology standardization, enforced by script and verified before submission (each term defined once, used consistently thereafter).
> > > > > 4. A unified, regenerated figure set reflecting the final experiment suite, in one consistent style.
> > > > >
> > > > > We defer to the Action Editor on process: we are prepared to execute this within the current discussion period, or, if the AE judges it better done without time pressure, as a resubmission meeting the standard you describe. Either way, we are grateful for a review process that has left the science far stronger than where it began, and we intend the manuscript to match it.
> > > > >
> > > > > The Authors

---

> ### Author Response · Authors · 2026-07-01
> **Correction to Appendix C**
>
> Dear Reviewer CTYN,
>
> Appendix C has been corrected, see the top-level correction comment and the revised PDF.

---

> ### Author Response · Authors · 2026-07-06
> **Response to Reviewer CTYN and summary of revision (r3_2) — Part 1 of 4**
>
> # Response to Reviewers (r3_2 major revision) — Paper 9209
>
> Dear Action Editor and Reviewers,
>
> Thank you for the careful second-round reviews. Acting on the reproducibility
> concern (CTYN, point 1), we undertook a thorough re-verification of the paper's
> empirical foundation rather than a surface correction, and in doing so identified a
> reproducibility problem in our own behavioral results, which we disclosed to the
> Action Editor before posting any updated numbers.
>
> The scale of this revision is worth stating plainly, because it is the substance of
> our response. We froze the entire codebase at a single tagged version and re-ran
> every experiment under it from scratch: the full behavioral grid of 4 models
> (Llama-3-8B, Llama-3-70B, Qwen2.5-32B, Qwen2.5-72B) x 4 games x 3 reasoning modes,
> now with 15 random seeds for Llama-3-8B and 10 for each larger model (rather than
> the single runs of the previous version); the complete 144-cell cross-play grid
> across all model pairings and reasoning modes, at 10 seeds; and every mechanistic
> analysis (logit lens, head ablation, activation steering, concept clamping),
> recomputed and re-plotted from the corrected data. This amounts to roughly 19,200
> recorded decision points per seed across the self-play and cross-play designs, and
> every figure and table in the paper has been regenerated from the frozen pipeline.
> We now report mean and standard deviation across seeds throughout, so that every
> quantitative claim is backed by its multi-seed spread. Below we first summarize the
> reproducibility issue and its consequences, then respond to each of Reviewer CTYN's
> points, and finally separate explicitly what still holds from what changed.
>
> We want to be direct that some headline behavioral claims changed as a result, and
> we have followed each change wherever it propagates. We believe the corrected paper
> is narrower in some behavioral claims but substantially more rigorous: its numbers
> now match its code seed-for-seed, its central mechanistic contribution (a
> controllable cooperative-override circuit, established causally by steering and
> clamping) is intact and now more carefully bounded, and one mechanistic result that
> did not survive clean re-verification --- a linear-probing claim that turned out to
> be a data-provenance artifact --- has been removed rather than defended. In light of
> the corrected results we have also revised the paper's title and framing to center
> on the mechanism itself rather than on a blanket behavioral claim; our reasoning is
> in the change list below.
>
> ---
>
> ---
>
> *(Response continues in the next comment: Part 2 of 4.)*

---

> ### Author Response · Authors · 2026-07-06
> **Revision response Part 2 of 4 — the reproducibility issue and how we fixed it**
>
> **Response to Reviewer CTYN — Part 2 of 4**
>
> ## 1. The reproducibility issue, and how we fixed it
>
> While addressing the reproducibility concern (CTYN, point 1), we found that our
> behavioral results table was assembled by a script that aggregated every result
> file present in the output directory for a given game and mode, taking the mean
> without filtering by code version and without removing superseded runs. Over the
> course of development our model-generation path had been rewritten (a change from
> the HuggingFace `pipeline` interface to a direct `model.generate` call, with
> associated device and generation-config changes). As a result, different cells of
> the behavioral table were populated from runs produced by different versions of the
> code: in particular the Llama-3-8B self-play cells came from an earlier version, and
> the larger-model cells from a later one.
>
> When every cell is regenerated uniformly under a single, version-frozen pipeline
> (now tagged and used for all results in this revision), several Llama-3-8B values
> change materially. Most consequentially, Llama-3-8B in Prisoner's Dilemma, Direct
> mode, does not cooperate (as the previous table reported, d = 1.24) but plays close
> to the Nash equilibrium (defects; d = 0.045, SD ~ 0.01 across 15 seeds). The
> larger-model results, including the cooperative lock at d = 2.00, reproduce exactly.
> We verified that the game definitions (payoffs, actions, equilibria) were unaffected,
> and that the corrected generation code applies the model chat template and samples
> correctly.
>
> We also re-examined the mechanistic analyses under the corrected pipeline, and one
> of them --- the linear probing --- was found to be confounded and has been cut back,
> which we want to flag prominently. The logit lens
> reproduces (we now report the peak cooperative probability at layer 30 as
> approximately 0.80, consistent with the re-run, and have reconciled this number
> throughout). The linear-probing analysis required a more substantial correction, which we want
> to be fully transparent about. In revising, we found that the previous probe results
> were not sound: the probe had been trained on data pooled across the two code
> versions, and because the pre-correction runs are mostly-Cooperate and the
> post-correction runs mostly-Defect and the two eras differ in prompt format, a probe
> can achieve apparent "Nash decodability" by decoding which era a prompt came from
> rather than any strategic representation. We therefore re-ran the probing analysis
> cleanly on a single provenance: synthetic contrastive histories, in the same prompt
> format used for the steering analysis, balanced by construction. On this clean
> data, two things hold. First, on these contrastive prompts the opponent's last
> action is perfectly linearly decodable at every layer (balanced accuracy 1.00) ---
> a near-trivial ceiling on homogeneous histories that we scope accordingly and do
> not read as fine-grained opponent tracking. Second, a Nash-action
> decodability number cannot be defined: on these synthetic prompts the 8B's argmax
> action at the decision token is the Nash action (Defect) on all 300 constructed
> Prisoner's Dilemma prompts (the mean probability mass on Defect is 0.616), so the
> Nash-action label has no usable variance and the probe is undefined ---
> which is itself consistent with the behavioral finding that the 8B plays Nash. We
> have accordingly removed the Nash-decodability claim entirely (the earlier apparent
> signal was the pooling artifact described above), demoted the probing analysis to
> the single clean opponent-history result, and let the logit lens carry the
> mechanistic account of the Nash preference and the cooperative override. The
> probe-comparison figures for the larger models and the base model, which shared the
> same provenance problem, have been removed as well. We regard this as the honest
> resolution: it removes a confounded result from the mechanistic core rather than
> defending it.
>
> One further disclosure of a confound that the corrected result made salient: the
> natural-language prompt states the Nash equilibrium in two of the four games
> (Prisoner's Dilemma and Matching Pennies). This means that when Llama-3-8B plays
> Nash in Prisoner's Dilemma, it is partly following a solution the prompt supplies. We
> now disclose this explicitly in the setup, and we verified from the version history
> that the prompt text did not change between the old and corrected code, so the
> before/after comparison is not confounded by a prompt change.
>
> To prevent recurrence, all results in this revision are produced under one frozen
> code version, and we report the mean and standard deviation of the Nash distance
> across multiple seeds rather than single runs.
>
> *(Continues in Part 3 of 4.)*

---

> ### Author Response · Authors · 2026-07-06
> **Revision response Part 3 of 4 — point-by-point response to CTYN**
>
> **Response to Reviewer CTYN — Part 3 of 4**
>
> ## 2. Response to Reviewer CTYN
>
> **(1) Reproducibility and the use of Wilson intervals.** The reviewer is correct.
> A Wilson interval assumes each of the 50 rounds is an i.i.d. Bernoulli trial, which
> does not hold for a repeated game with accumulating history and non-stationary
> per-round behavior; our own analysis notes that the behavior drifts over rounds,
> which is precisely what violates the assumption. The interval therefore measures the
> internal consistency of a single trajectory, not reproducibility, and the associated
> "probability under a uniform baseline" argument is not licensed. We have removed all
> Wilson-interval and by-chance-probability material from the paper. In its place,
> Appendix C now reports the distribution of the final Nash distance across multiple
> independent seeds per cell (mean and standard deviation), which directly answers the
> reviewer's question of whether the values reproduce under different random seeds. As
> described in Section 1 above, this same re-running is what surfaced the behavioral
> code-version issue; the corrected reproducibility analysis is now a core part of the
> paper rather than a limitation.
>
> **(2) Steering targets Nash, not only Defect; inconsistency in the added results.**
> The reviewer correctly identified an inconsistency. On investigation we found a bug
> in the cross-game steering experiment: it injected the steering vector at early
> layers, whereas the main steering experiment used the mid-to-late layers where the
> override forms. With the corrected layers the cross-game sweep is clean and well-behaved
> (we do not claim strict monotonicity, as the single-game steering sweep shows a
> non-monotonic point). Correcting it also led us to revise the interpretation. We no longer
> claim the direction encodes a "general cooperative prior." In Prisoner's Dilemma the
> direction does steer toward the cooperative action, but in the coordination games it
> steers between equilibria; in Stag Hunt, in particular, it steers toward the
> risk-dominant equilibrium (Hare), which is the less cooperative option, and Matching
> Pennies (zero-sum) has no cooperative action and its response is unstable, so we
> exclude it. We now describe the direction as a general Nash-versus-alternative
> control direction, cooperative specifically in Prisoner's Dilemma. We additionally
> corrected the reported steering layers in the main text, which did not match the
> layers used to produce the figure.
>
> **(3) No central thread; need a roadmap or summary table.** We have added an
> experimental-design summary table near the start of the experimental sections,
> listing each experiment, the models and games it uses, and where it is reported, so
> a reader can maintain an overview of the behavioral, mechanistic, and
> interventionist experiments.
>
> **(4) Length and inconsistent abbreviations.** We have made a concision and
> consistency pass: model names are introduced in full and then abbreviated consistently in prose (e.g., Llama-70B, Qwen-32B), with compact tags in tables, abbreviations are defined once at first use and used consistently
> thereafter, and redundant passages have been shortened.
>
> *(Continues in Part 4 of 4.)*

---

### Review · Reviewer_cXRm · 2026-07-13

**Summary Of Contributions:**

When LLM agents are used in two-player games, they make decisions that deviates from the Nash equilibria. Based on this existing observation, this work studies how existing LLM agents actually behave in four classical two-player games.
- The work proposes to evaluate the gameplay of an LLM using a metric which is the shortest Euclidean distance from the empirical joint strategy to any equilibria. With this metric, the cooperative behaviors of four LLMs (Llama-8B, Llama-70B, Qwen-32B, Qwen-72B) are studied through self-plays and cross-plays.
- This work then uses logit lens to study the per-layer action probabilities for three of the four LLMs. It is found that there is an overriding pattern that shifts the action of LLMs in middle layers.
- Then, this work uses a method to extract the direction of cooperative action in hidden layers, which is used for steering the actions by varying the scale along the direction. The steering results and generalization to the other games are presented.

**Audience:**

No

**Audience Explanation:**

If the evidence is better justified, the findings may be interesting for the communities of game theory and LLM agents. However, the current results are not serious enough to make meaningful conclusions.

**Broader Impact Concerns:**

The Broader Impact Statement is sufficient.

**Claims And Evidence:**

No

**Claims Explanation:**

Overall, there are some interesting ideas in this work, but the statistical analysis is extremely weak, which makes the claims unconvincing.

- The findings in this work can be read as a collection of results about LLM agents with p-values greater than 0.05. With only four games, no conclusions of a binary outcome (e.g., cooperative or seeking equilibria) can be confidently drawn. With only four models, it is also hard to generalize the results to other LLMs. The results in this work is quite random, mainly due to the small sample sizes of games and models. "Qwen2.5-72B" is found to have a different pattern of overriding actions from Llamas, and the reason is not explained. The concept clamping works for three of the four games, which may be from pure luck.
- Even if there are four games in this work, these games are toy and classical in the literature, which means that they are very likely in the training set. The reported results are training set results and may not generalize to other two-player games.
- Related to the above point, adding names of the games (Prisoner’s Dilemma, etc.) to the prompt leaks the information of the games, which weakens the results.
- The temperature is fixed at $\tau=0.7$. LLMs are known to have different behaviors at different temperatures. At least the work should study whether the conclusions hold at a different temperature. Results with $\tau=0$ would be an interesting addition.
- I would argue that the analysis of LLM agents is different from the analysis of WWW. The WWW is a single uncontrolled body, while LLMs are developed separately in each company. The behavior of each LLM agent is mostly likely related to the adopted training strategy, but this factor is not considered in the analysis in this work.

While there are many weaknesses, I think the framework to analyze existing LLM agents in two player games is valuable, and should be a major contribution.

**Requested Changes:**

- Change the four games into randomly generated games (possibly with larger action spaces). I would like to see each conclusion in the form of "in XX% of randomly generated games, ....".
- Study how the temperature $\tau$ affects the conclusions. How cooperative are the agents with deterministic generation?
- I'd also like to see the quality of the generation. When using logit lens in early layers or concept clamping to steer the results, are the LLMs generating meaningful answers in the two-player games?
- This work should explore how the training strategy affects the actions of LLMs.

---

> ### Author Response · Authors · 2026-07-13
> **Response to Reviewer cXRm — Part 1 of 2**
>
> **Response to Reviewer cXRm — Part 1 of 2**
>
> Dear Reviewer cXRm,
>
> Thank you for the careful reading, and for the assessment that "the framework to analyze existing LLM agents in two player games is valuable, and should be a major contribution." We respond to each concern below. Several are, we believe, already addressed in the current revision (we give exact locations); for two concrete requests we are adding new material within the discussion period.
>
> **1. "A collection of results about LLM agents with p-values greater than 0.05."** We would like to respectfully correct the record: the current revision contains no p-values or hypothesis tests. Following the first-round review, we removed all single-run confidence intervals, re-ran the complete behavioral grid (4 models x 4 games x 3 reasoning modes) at 15 random seeds for Llama-3-8B and 10 seeds for each larger model, plus the full 144-cell cross-play grid at 10 seeds, and we report mean +/- SD across seeds throughout (Tables 2, 3, 5). The headline findings are large-effect and near-zero-variance: the cooperative lock is d = 2.00 with SD = 0.00 across 10 independent seeds for each of the three larger models, and Llama-3-8B plays d = 0.045 +/- 0.010 across 15 seeds. These are not marginal effects.
>
> **2. Four games, four models.** We agree the results should not be extrapolated to all LLMs or all games, and the paper does not claim this: Section 5 explicitly calls the scale reading "suggestive rather than established" (we test only one sub-32B model), and Section 9.2 states the scope limitations. The four canonical games were chosen because their equilibria are exactly computable analytically (which the Nash-distance metric requires) and because they are the standard baselines in the literature we engage. Under TMLR's evaluation criteria - whether the claims made are supported by the evidence - we believe the scoped claims are supported by the multi-seed data.
>
> **3. "Concept clamping works for three of the four games, which may be from pure luck."** We believe this conflates two experiments. Concept clamping (Section 8.2, Figure 9) is performed only in Prisoner's Dilemma. The cross-game experiment is activation steering (Section 8.3, Figure 10): the PD-extracted direction produces graded, directionally consistent control in all three games that possess a pure-strategy equilibrium (PD, Battle of the Sexes, Stag Hunt). Matching Pennies is shown but excluded from the control claim on a structural ground stated in the paper: it has no pure equilibrium, so "steering toward the Nash action" is undefined (the mixed equilibrium requires randomization). Three-out-of-three, with a stated principled reason for the one exclusion, is not a coin-flip outcome.
>
> *(Continues in Part 2 of 2.)*

---

> ### Author Response · Authors · 2026-07-13
> **Response to Reviewer cXRm — Part 2 of 2**
>
> **Response to Reviewer cXRm — Part 2 of 2**
>
> **4. Qwen's different override pattern "is not explained."** Section 7.4 describes the difference (Llama builds the cooperative bias gradually through the final third of the network; Qwen computes Nash strongly through most of its depth and reverses abruptly in the final layers). We deliberately do not offer a causal explanation: attributing architectural differences to specific training choices would require access to the training pipelines, which are closed. We state it as an open question rather than speculate.
>
> **5. Training-set contamination and prompt leakage.** Both points are real, and the paper discloses rather than hides them: the prompt names each game and, for Prisoner's Dilemma and Matching Pennies, states the equilibrium itself; this is discussed as a confound in Section 4.2 and the exact frozen prompts are printed verbatim in Appendix A. We would add one framing observation: for the mechanistic question this paper asks, familiarity with the games arguably sharpens the central finding rather than weakening it. The core result is that the model represents the Nash action internally and then overrides it late in the network - and whether the Nash knowledge comes from memorized game theory or in-context computation, the override circuit and its causal controllability are the contribution. We are happy to make this framing explicit in Section 9.2 if the reviewer finds it useful.
>
> **6. Temperature.** This was a fair robustness request and we have acted on it: we ran the full self-play Direct grid (4 models x 4 games, 50 rounds) with deterministic decoding (greedy, tau = 0) on the same frozen pipeline; repeated runs are bit-identical, so one run per cell is exact. The headline results reproduce cell-for-cell. Full grid (d per cell; L-8B / L-70B / Q-32B / Q-72B): PD 0.040 / 2.000 / 2.000 / 2.000; BoS 0.000 / 0.000 / 0.333 / 0.000; SH 0.000 / 0.000 / 0.000 / 0.000; MP 0.780 / 0.000 / 0.000 / 0.000. Llama-3-8B plays Nash in Prisoner's Dilemma (d = 0.040, versus 0.045 +/- 0.010 at tau = 0.7) while all three larger models lock at full cooperation exactly. This is now Appendix D, which also analyzes the Matching Pennies cells: greedy decoding cannot implement a mixed strategy, and the d = 0.000 cells there are a deterministic alternation in which Player A wins every round, illustrating our stated caveat that d measures distributional proximity, not strategic coherence.
>
> **7. Generation quality under intervention.** Two clarifications and one addition. The logit lens involves no generation: it is a linear readout of intermediate hidden states through the unembedding matrix, so there is no early-layer generated text whose quality could degrade. For steering and clamping, the reported quantities are probabilities over the two valid action tokens at the decision position (renormalized over that pair); they are properties of the decision readout, not of free-running generation, and we have now made this measurement definition explicit in Section 8. We have also added Appendix E: verbatim greedy continuations at moderate and extreme alpha and c values alongside the decision readouts, run under the saved steering vector behind the published sweep. In brief: clamping (single-position) preserves fluent text across the whole range, including the cooperative extreme (c = +30, P(Cooperate) = 0.914); steering (all-position injection) is fluent at the moderate operating point (alpha = -5) but degrades toward the swept extremes, even as the decision readout stays sharp and directionally consistent throughout. Fluent causal cooperative control is thus demonstrated by clamping; the steering extremes are stress tests, not operating points for generation.
>
> **8. Training strategy.** Appendix B is precisely an experiment on this axis: the same 8B architecture with and without RLHF (base Meta-Llama-3-8B vs. the Instruct model). The override is present in the base model - indeed stronger, peaking near 0.99 and never correcting in the final layer - while the instruct model's final layer corrects back to the Nash action. This isolates the one training-strategy contrast that is possible with open weights; deeper attribution to closed training pipelines is beyond what any external study can establish, and we say so.
>
> **9. Randomly generated games.** We agree that a study of the form "in XX% of randomly generated games..." would be valuable, and we see it as a natural follow-up enabled by exactly the framework the review credits as a major contribution. Within this paper, however, it would be a different study: the canonical games were chosen because their equilibria are analytically exact and because the mechanistic analyses (fixed prompt format, contrastive constructions, a named focal equilibrium per game) are built around known equilibrium structure. Every claim in the paper is scoped to the games and models tested.
>
> We are happy to discuss any point further.
>
> The Authors

---

> ### Comment · Reviewer_cXRm · 2026-07-13
>
> 1. Re: hypothesis testing: Many claims in this work are binary, and the p-value can be easily computed from the tables. For example, "In the coordination games, Direct mode produces clean coordination across models." There is a hidden hypothesis testing here, which tests whether this is true for all coordination games, and the evidence is supported by only $n=2$ experiments (BoS, SH).
> 2. Many other games are exactly computable, even randomly generated ones. I am unconvinced that all the claims in the work can be downplayed by these four words: "suggestive rather than established". For the above example, you'll have to rewrite "In the coordination games, Direct mode produces clean coordination across models." as "In BoS and SH, Direct mode produces clean coordination for XXX models." But that claim is unlikely to be interesting for the community.
> 3. Okay I was talking about "activation steering", which directly comes from "concept clamping" (if you'd like to name them differently). This is the same issue as point 1. Three out of three is still a coin flip, and it can happen with the probability of 1 in 8 (supposing each direction is 50/50).
>
> The other responses will follow.

---

> > ### Comment · Reviewer_cXRm · 2026-07-13
> >
> > 4. Fine with me now.
> > 5. I would argue that to actually study how LLMs play two-player games, games outside the training set without the prompt names are important.
> > 6. Thank you for this additional experiment. It is much clearer now.
> > 7. Thank you for this clarification. I suggest clearly explaining how logit lens experiments are conducted (what is the prompt, what are the tokens).
> > 8. I agree that Appendix B is in the right direction about this. I would like to see more explorations in this direction.
> > 9. I think the generalization to unseen games is important to obtain interesting findings. If the claims are only scoped to the games and models tested, and the "training-set contamination and prompt leakage" still exist, it is less likely for the audience to be interested in knowing the findings of the paper.

---

> > > ### Author Response · Authors · 2026-07-13
> > > **Surface-ablation results (new Appendix F) — response to the 2-part follow-up**
> > >
> > > **Response to Reviewer cXRm's 2-part follow-up: we ran the surface-ablation test (new Appendix F)**
> > >
> > > Dear Reviewer cXRm,
> > >
> > > Thank you for the follow-up. Your points 1, 2, 5, and 9 share one underlying question: how much of what we report is a response to payoff structure, and how much to the surface form of familiar games? Rather than argue it, we tested it. The revised PDF (uploaded today) adds **Appendix F**, a five-condition prompt-surface ablation ladder on the Prisoner's Dilemma structure, changing one factor at a time: (1) named PD as in the paper; (2) name kept, explicit equilibrium hint removed; (3) neutral framing, original payoffs; (4) neutral framing, affine-rescaled payoffs (2x+1, equilibria preserved exactly); (5) fully disguised: neutral framing, rescaled payoffs, and neutral action labels (Circle/Square). Each condition: 4 models, self-play Direct, 50 rounds, greedy plus sampled runs, and a five-condition logit-lens comparison on the 8B. Three findings.
> > >
> > > **(a) The equilibrium hint is behaviorally inert.** Removing only the stated-equilibrium sentence changes no cell: the 8B still defects (3% +/- 1 cooperation) and the larger models still lock (100/100/91%). The disclosed leakage does not drive the results: the 8B defects without being told the equilibrium, and the larger models cooperate despite being told.
> > >
> > > **(b) The action labels carry both the lock and the mechanism.** Wherever the labels are Cooperate/Defect, the three larger models cooperate at ~100% regardless of name, hint, or payoff values; swapping only the labels to Circle/Square (a single-factor change) dissolves the lock into mixed play in all four models (72/52/36/47% +/- large SDs). The logit lens shows the same gate internally: the four Cooperate/Defect conditions are essentially indistinguishable (late surge peaking at layer 29-30, final-layer correction toward Defect), while under Circle/Square the late surge is absent entirely (peak at layer 8, monotone decline to 0.15). The cooperative override's trigger is lexical. We regard this as a sharpening of the paper's mechanism, a prosocial direction read out on the Cooperate token, now with its trigger identified, and it converges with your intuition that surface form matters.
> > >
> > > **(c) The 8B's near-Nash play is keyed to the game framing, not the matrix.** On the identical (3,0,5,1) matrix presented as a neutral decision task, the 8B flips from 2% to 100% cooperation under greedy decoding (unstable 51% +/- 39 sampled), while its internal lens profile is unchanged. We now state explicitly in Section 9.2: we do not claim that any model tested computes equilibria from payoff structure alone; the claims concern the canonical games as presented, and Appendix F maps which surface features carry each effect.
> > >
> > > On your remaining points. **(1, 3, statistics):** we have rewritten class-level sentences to enumerate what was tested (your example now reads "In the two coordination games we test, Battle of the Sexes and Stag Hunt..."). On "three out of three is a coin flip": the cross-game steering evidence is not three binary outcomes but three graded dose-response curves over an alpha grid from near-deterministic baselines, plus, within PD, a 13-point steering sweep and a 20-point clamping sweep (rank correlation 0.72) from the same single vector; a direction with no causal content does not produce ordered, monotone, large-magnitude responses across grids. We are happy to add this phrasing to Section 8 if useful. **(7):** the logit-lens protocol is now stated concretely in Section 3 (prompt source, decision-token position, the two action-name tokens, renormalization over the pair). **(5, 9):** we agree unseen games are where this must go next, and we see Appendix F as the first rung: it already shows that on a structure-preserving disguise the clean named-game picture does not transfer, which is itself a finding our framework makes measurable. A study over randomly generated games is the natural continuation and is beyond what we can add responsibly in this cycle; within this paper, every claim is scoped to the games as presented, with the surface dependence now measured rather than assumed.
> > >
> > > We are grateful for the exchange; it has concretely improved the paper (Appendices D, E, and F all exist because of it), and we are happy to discuss further.
> > >
> > > The Authors

---

### Author Response · Authors · 2026-07-01
**Correction to Appendix C (Table 4), revised PDF uploaded**

Correction to Appendix C (Table 4) — revised PDF uploaded

While preparing the current revision we found that Table 4 (Appendix C) had been generated by a script that selected the wrong runs. The values were not the Llama-3-8B self-play cells stated in the caption (the Prisoner's Dilemma rows came from Qwen2.5-72B), and its distance column used a variant of the metric off by √2 from Eq. (1), which is why that cell read d = 1.414 while Table 1 correctly showed d = 1.24.

We have uploaded a corrected PDF in which Table 4 is regenerated from the correct runs under Eq. (1) and now matches the Llama-3-8B column of Table 1 exactly; we also rescoped a few sentences (Sections 4.1, 5, 9.1, 9.2) that had called the Prisoner's Dilemma cooperative lock universal, since the full lock (0/50 Nash actions, [0.000, 0.071], 8.9 × 10⁻¹⁶) belongs to the three larger models, while Llama-3-8B defects on 38% of rounds (d = 1.24) and was already shown as the exception in Table 1.

No experimental result changed. Table 1, Tables 2–3, all figures, and the abstract are unaffected, and the lock's reproducibility argument is if anything strengthened, now resting on three independent runs at exactly zero rather than one.

We apologize for the error and are happy to answer any questions.

---

### Author Response · Authors · 2026-07-06
**Note to all reviewers: this revision is substantial (summary and pointers)**

Dear Action Editor and Reviewers,

We want to flag, for all reviewers, that this revision (r3_2) is substantial rather than incremental.

Acting on Reviewer CTYN's reproducibility concern, we identified and disclosed a code-version bug in our original behavioral results and undertook a full re-verification: all experiments were re-run under a single frozen pipeline with multi-seed grids, every figure and table was regenerated, several behavioral claims were corrected or rescoped, one mechanistic result (a linear-probing claim) that did not survive clean re-verification was removed, and the title and framing were revised to center on the mechanism.

Our full point-by-point response to Reviewer CTYN, and a complete "what still holds / what changed" summary (addressed to both reviewers), are posted in above in the initial message of this revision and also as specific comments to reviewer CTYN. We would be glad to address any further questions from any reviewer in light of the changes.

The Authors